# Combined chemical genetics and data-driven bioinformatics approach identifies receptor tyrosine kinase inhibitors as host-directed antimicrobials

Cornelis J. Korbee [1], Matthias T. Heemskerk[1], Dragi Kocev[2], Elisabeth van Strijen[1], Omid Rabiee[1], Kees L.M.C. Franken[1], Louis Wilson[1], Nigel D.L. Savage[1], Sašo Džeroski[2], Mariëlle C. Haks[1] & Tom H.M. Ottenhoff[1]

Antibiotic resistance poses rapidly increasing global problems in combatting multidrug-resistant (MDR) infectious diseases like MDR tuberculosis, prompting for novel approaches including host-directed therapies (HDT). Intracellular pathogens like *Salmonellae* and *Mycobacterium tuberculosis* (*Mtb*) exploit host pathways to survive. Only very few HDT compounds targeting host pathways are currently known. In a library of pharmacologically active compounds (LOPAC)-based drug-repurposing screen, we identify multiple compounds, which target receptor tyrosine kinases (RTKs) and inhibit intracellular *Mtb* and Salmonellae more potently than currently known HDT compounds. By developing a data-driven in silico model based on confirmed targets from public databases, we successfully predict additional efficacious HDT compounds. These compounds target host RTK signaling and inhibit intracellular (MDR) *Mtb*. A complementary human kinome siRNA screen independently confirms the role of RTK signaling and kinases (BLK, ABL1, and NTRK1) in host control of *Mtb*. These approaches validate RTK signaling as a drugable host pathway for HDT against intracellular bacteria.

[1] Department of Infectious Diseases, Leiden University Medical Center, Albinusdreef 2, Leiden 2333 ZA, The Netherlands. [2] Department of Knowledge Technologies, Jožef Stefan Institute, Jamova Cesta 39, Ljubljana 1000, Slovenia. Cornelis J. Korbee, Matthias T. Heemskerk, and Dragi Kocev contributed equally to this work. Mariëlle C. Haks and Tom H.M. Ottenhoff jointly supervised this work. Correspondence and requests for materials should be addressed to M.C.H. (email: m.c.haks@lumc.nl) or to T.H.M.O. (email: t.h.m.ottenhoff@lumc.nl)

With an estimated 1/4 of the world population carrying a latent *Mycobacterium tuberculosis* (Mtb) infection, 10.5 million new cases, and 1.8 million deaths annually, tuberculosis (TB) is an increasing global health issue[1–3]. This is further aggravated by the emergence of multi-, extensively, and totally drug-resistant (MDR/XDR/TDR) *Mtb* strains, threatening to render TB untreatable using current antibiotics[4–6]. In 2015, 480,000 patients suffered from MDR-TB.

Although novel candidate antibiotics have recently been identified[7], current antibiotics already cover the majority of drugable targets of pathogens, resulting in a continuous decline in the number of new and approved antibiotics[8–13]. Intracellular bacteria such as *Salmonellae* and *Mtb* pose additional challenges by manipulating host-signaling pathways to subvert innate and adaptive immunity. This, however, also creates potential for novel treatment strategies like host-directed therapy (HDT), to reprogram the host immune system by pharmacological and chemical–genetic manipulation. Importantly, HDT-driven manipulation of host-signaling pathways may be effective also against drug-resistant bacteria, and helps to restore host control of infection in metabolically perturbed cells[14, 15]. Several recent studies, including our own, have demonstrated the feasibility of HDT approaches to inhibit bacteria both in vitro in human and murine cells[16–20] and in vivo in mice, rabbits, and zebrafish[21–31]. Using reciprocal chemical genetics targeting the human kinome, we previously identified AKT1 as a central regulator of *Salmonella enterica* serovar Typhimurium (Stm)-, *Mtb*-, and MDR-*Mtb* survival. Treatment of infected cells with the kinase inhibitor H-89 significantly decreased intracellular bacterial loads. Despite H-89 being known as a PKA inhibitor, we demonstrated that this compound inhibited intracellular bacteria by targeting AKT1[16]. However, H-89 had a substantially lower impact on intracellular growth of *Mtb* compared to *Stm*, suggesting that *Mtb* modulates additional host-signaling pathways to survive. This is in agreement with reports that *Mtb* arrests vesicle maturation at an earlier stage than *Stm*[16, 32, 33]. Other studies identified additional drugable human kinases that regulate *Mtb* survival, including TGFβRI and CSNK1[18] and imatinib-sensitive kinases ABL1 and ABL2[21]. In addition to kinases and kinase inhibitors, other potential targets and compounds for TB HDT were identified, including two antipsychotics (haloperidol and prochlorperazine) and an antidepressant (nortriptiline)[19], phosphodiesterase inhibitors[22, 23], anti-inflammatory agents like ibuprofen[25], the FDA-approved drug zileuton[26], the antidiabetic drug metformin[34], phenylbutyrate[35, 36], and human metabolic targets[37, 38]. Nevertheless, the field of TB HDT has not fully progressed toward clinical application and many interactions between the host and bacterium remain to be deciphered. Therefore, better compounds are urgently needed as drug candidates for TB HDT, as well as for the identification of cellular events occurring at the host–pathogen interface, which may enable rational drug design for HDT.

There are several major challenges to be overcome to facilitate host-directed chemical–genetic studies targeting intracellular pathogens, particularly studies that aim at discovering key host pathways manipulated by *Mtb*. First, it is extremely difficult to generate sufficient quantities of primary macrophages (Mφs), the natural target cells for *Mtb* infection, from human donors for medium-throughput screens, even by pheresis. Second, the often-used THP-1 monocytic cell line requires PMA stimulation for differentiation, which massively affects cell signaling and vesicular trafficking[39, 40], thus confounding cellular signaling studies. Third, there is a lack of fast (compared to the classical 3-week *Mtb* colony-forming unit (CFU) assay), robust, and widely applicable readouts for rapid screening. Finally, achieving stable genetic knockdown in human primary macrophages is

challenging, especially in large siRNA screens where knockdown efficiency of each individual gene cannot broadly be confirmed. To solve these problems, we have developed a rapid, medium-throughput, fluorescence-based screening assay to determine bacterial load by automated flow cytometry in the highly manipulable human HeLa and MelJuSo cell lines infected with (myco)bacteria expressing novel (myco)bacterial fluorescent protein constructs. Our identification of the MelJuSo cell line as a novel *Mtb* infection model has several important advantages: MelJuSo cells are suited to large-scale screening assays as they are more homogeneous than primary cells, do not require additional stimuli like PMA for maturation, can be efficiently manipulated using RNAi, and can be infected by mycobacteria[41]. We have shown in the past that human melanocytes can efficiently present mycobacterial antigens to HLA class II-restricted CD4 T cells[42] and have successfully used MelJuSo to dissect molecular pathways of MHC class II presentation in human cells[43, 44]. Our novel fluorescence-based bacterial growth assay is applicable for both siRNA and chemical compound screens, and is equally suitable for both *Stm* and *Mtb* despite the vast differences in their intracellular "lifestyles" and replication rates (20 min and 18 h, respectively)[6, 32, 45, 46], demonstrating the versatility of this assay.

We used this screening assay in drug-repurposing screens, and identified compounds with host-directed anti-(myco)bacterial activity against *Mtb* and *Stm*, outperforming published HDT compounds' activities. Based on these data, together with confirmed target profiles of the screened compounds, we next developed a predictive in silico model in order to be able to identify additional HDT compounds. This model was applied to predict host-directed compounds among all compounds present in the PubChem repository and to identify their key targets with predicted activity against intracellular *Stm* or *Mtb*. Interestingly, both our experimental wet lab screens and the novel in silico predictive model identified inhibitors of (growth factor) receptor tyrosine kinases (RTKs) and downstream intermediates of RTK signaling as candidate host-directed drugs to control intracellular infection. Moreover, an siRNA screen of the human kinome in *Mtb*-infected human cells independently validated a key role for RTK signaling in host control of *Mtb*. Thus, using two independent chemical–genetic experimental approaches as well as a computational method, we find and validate RTK signaling as a novel important host pathway that controls intracellular *Mtb* (including MDR-*Mtb*) survival. This pathway is druggable by compounds and drugs including clinical drugs such as dovitinib, AT9283, and ENMD-2076. These findings offer new approaches to combat intracellular infectious diseases in the face of rapidly rising multidrug resistance.

## Results

**Flow cytometry-based assay for intracellular bacterial load**. To allow identification of host-directed drugs and host pathways controlling intracellular bacterial survival, we developed a fast, robust, and novel assay suited for medium-throughput (96-well) compound and siRNA screening. Flow cytometry was used to assay intracellular bacterial load using fluorescent strains of *Stm* and *Mtb*. For initial reference purposes, we tested the kinase inhibitor H-89, which we had used in previous work[16] (Supplementary information, Supplementary Figures 1–4). As mentioned, after screening several cell lines, we identified the MelJuSo cell line as a novel *Mtb* infection model. To first validate the MelJuSo model system, we tested already-published host-directed compounds with known activity against *Mtb* here in this model; indeed, as expected, these known compounds also reduced *Mtb* loads in MelJuSo cells and in human macrophages upon short (overnight) treatment (at standard 10 μM concentrations).

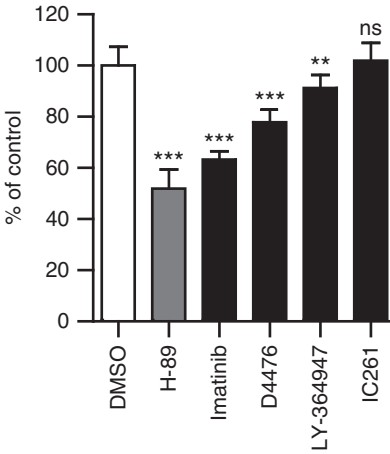

**Fig. 1** Verification of host-directed *Mtb* inhibitors from literature in the novel screening assay. Overnight treatment of MelJuSo cells infected with *Mtb* constitutively expressing stable DsRed with compounds that were previously reported to be active against *Mtb* at 10 µM or with DMSO at equal v/v. H-89 is used as a positive control at 10 µM[16]. The *Mtb* bacterial load is displayed as a percentage of the DMSO control ± standard deviation to indicate the extent of bacterial inhibition. Statistically significant difference compared to DMSO was tested using a one-way ANOVA. (ns not significant, **p value < 0.01, ***p value < 0.001)

Supplementary Figure 1 describes the assay in more detail. As shown in Fig. 1, imatinib, D4476, and LY-364947 all decreased the *Mtb* bacterial load in our model, which was well in agreement with previous studies[18, 21]. In addition, we also confirmed that a dual inhibitor of TGFbetaR1 and casein kinase 1 (D4476) inhibited *Mtb* more potently than the individual inhibitors of TGFbetaR1 (LY-364947) and casein kinase 1 (IC261) alone, confirming data from Jayaswal et al.[18] (Fig. 1). Finally, during the work described in this paper also, haloperidol[19] was confirmed to inhibit both *Stm* and *Mtb* in our model (Table 1) and these results will be discussed in more detail below. Thus, the results obtained in our novel MelJuSo-*Mtb* infection model and flow cytometry-based readout of intracellular infection faithfully reproduce the reported inhibitory effects of previously published compounds, providing important biological plausibility for the system.

Of note, none of the published compounds evaluated above was more potent in inhibiting intracellular *Mtb* than our reference compound H-89. We therefore used our assay to screen chemical libraries to identify host-directed inhibitors with more potent activity than H-89.

**Identification of host-directed antimicrobial compounds**. We next applied the novel screening assay for a TB drug-repurposing screen of a library of 1260 pharmacologically active compounds (LOPAC) in order to identify host-directed compounds with stronger activity against intracellular *Mtb* than H-89. The primary screen in the MelJuSo-*Mtb* intracellular infection model

**Table 1 Details of validated hit compounds from the *Mtb* and *Stm* LOPAC screens**

| Abbr. | Compound name | Alternative name(s) | Primary sceen z-score | Rescreen z-score | Activity |
|---|---|---|---|---|---|
| *Mycobacterium tuberculosis* | | | | | |
| **SU** | SU 6656 | 2,3-Dihydro-*N,N*-dimethyl-2-oxo-3-[(4,5,6,7-tetrahydro-1H-indol-2-yl)methylene]-1H-indole-5-sulfonamide | **−5.79** | **−10.51** | Src family kinase inhibitor |
| **Q** | Quinacrine dihydrochloride | | **−5.25** | **−9.90** | MAO inhibitor |
| **SB** | SB 216763 | 3-(2,4-Dichlorophenyl)-4-(1-methyl-1H-indol-3-yl)-1H-pyrrole-2,5-dione | **−6.02** | **−8.29** | GSK-3 kinase inhibitor |
| **G** | GW5074 | 3-(3, 5-Dibromo-4-hydroxybenzylidine-5-iodo-1,3-dihydro-indol-2-one) | **−4.86** | **−6.98** | Raf1 kinase inhibitor |
| **T494** | Tyrphostin AG 494 | N-Phenyl-3,4-dihydroxybenzylidene cyanoacetamide | **−3.83** | **−6.93** | EGFR kinase inhibitor |
| **L** | 3′,4′-Dichlorobenzamil hydrochloride | L-594,881 | **−3.87** | −5.13 | $Na^+/Ca^{2+}$ exchanger inhibitor |
| **H** | Haloperidol | | **−3.77** | −2.96 | D2/D1 dopamine receptor antagonist |
| *Salmonella typhimurium* | | | | | |
| **T** | Trimethoprim | | **−4.06** | −12.18 | Antibiotic; dihydrofolate reductase inhibitor |
| **H** | Haloperidol | | **−3.90** | −12.09 | D2/D1 dopamine receptor antagonist |
| **M** | Mibefradil dihydrochloride | Ro 40-5967; (1S,2S)-2-[2[[3-(2-benzimidazolylpropyl]methylamino]ethyl]-6-fluoro-1,2,3,4-tetrahydro-1-isopropyl-2-naphthyl methoxyacetate dihydrochloride | **−3.64** | **−12.76** | $Ca^{2+}$ channel blocker |
| **O** | Ofloxacin | Ofloxacine; DL-8280; HOE-280 | **−3.45** | −11.60 | Antibiotic; DNA synthesis inhibitor |

*Z*-scores lower than the *z*-score of H-89 are displayed in bold

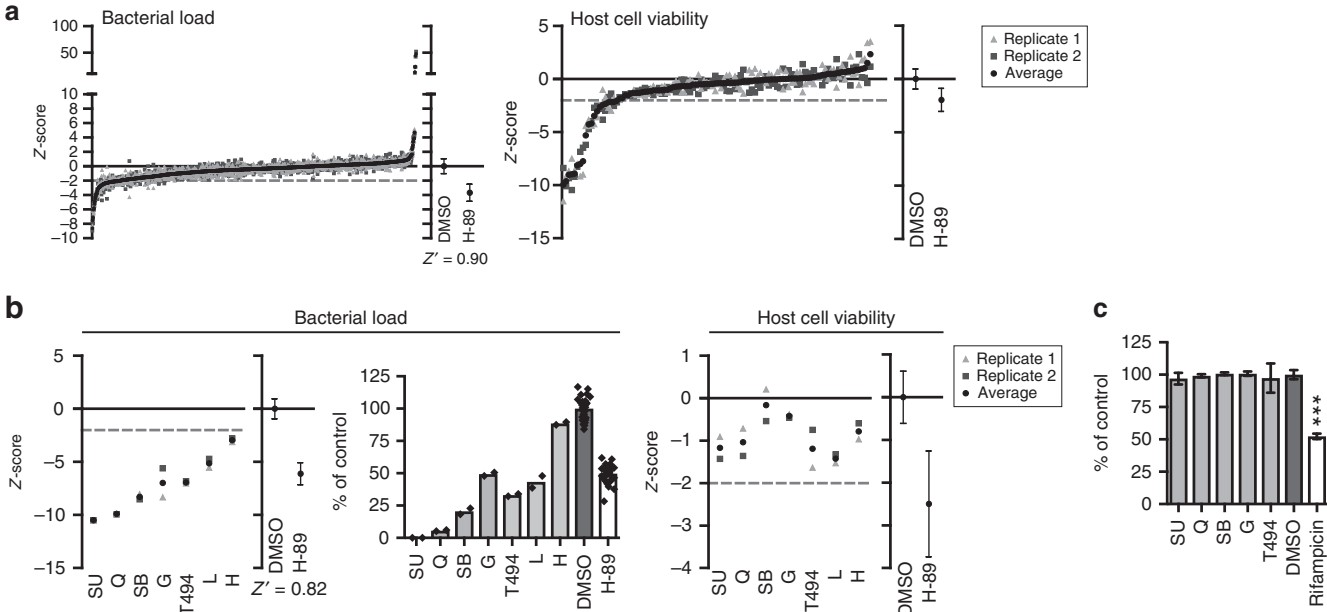

**Fig. 2** Identification of host-directed compounds inhibiting *Mtb*. **a** Results of a screen of 1260 compounds of the LOPAC library at 10 µM in the MelJuSo-*Mtb* infection model using *Mtb* constitutively expressing stable DsRed, expressed as *z*-scores (left panel). Individual replicates of the screened compounds are shown as gray points and the average *z*-score for each compound is displayed in black. The average *z*-score and standard deviation of the controls (DMSO and H-89) are displayed separately and the assay window expressed as a Z'-factor is shown below. Cell viability *z*-scores of the 110 hit compounds are shown in the right panel. The dashed line depicts a cutoff at a *z*-score of −2. **b** A rescreen of the hit compounds that were superior to H-89 without affecting cell viability at 10 µM is shown as in **a**. The bacterial load is expressed as a *z*-score in the utmost left panel and as a percentage of control value in the middle panel to indicate the extent of bacterial inhibition. Individual screening datapoints are overlaid on the bar graph. Compound abbreviations: SU = SU 6656, Q = quinacrine, SB = SB 216763, G = GW5074, T494 = tyrphostin AG 494, L = L-594,881, H = haloperidol. **c** *Mtb* broth culture treated for 6 days with the five hit compounds of the *Mtb* screen at 10 µM. Rifampicin (20 µg/ml) was used as a positive control. The average bacterial density ± standard deviation of four replicates from a representative experiment (out of three experiments) is shown, expressed as a percentage of the DMSO control. Statistically significant difference compared to DMSO was tested using a one-way ANOVA ($F_{(6,25)} = 81.66$; ***$p$ value < 0.001)

identified 110 compounds that significantly reduced and 16 compounds that increased intracellular bacterial loads. Ninety of these did not affect host cell viability (Fig. 2a and Supplementary Table 2) and were therefore pursued further. Seven compounds decreased *Mtb* bacterial load more potently than H-89 (Table 1). A rescreen of these seven compounds confirmed their activity and five of these compounds again surpassed H-89 (SU 6656, quinacrine, SB 216763, GW5074, and tyrphostin AG 494; Fig. 2b and Table 1). Figure 2b shows *z*-score values in the left panel, with the actual percentage inhibition of *Mtb* growth shown in the middle panel, expressed as the % of control value. These latter data confirmed the strong inhibitory effect of these HDT compounds on intracellular *Mtb*. We next confirmed that these compounds exerted their antimicrobial effects via the host by excluding any direct microbicidal activity against extracellular *Mtb* (Fig. 2c). As a control, the classical Mtb antibiotic rifampicin significantly inhibited *Mtb*.

To investigate whether compounds also existed with host-directed activity against *Stm*, and whether their activity was selective for *Mtb*, *Stm*, or both, we also screened the same LOPAC library using the very similar HeLa-*Stm* infection model (Fig. 3a). Twelve compounds were identified that significantly reduced the *Stm* bacterial load and 10 of these did not affect host cell viability (Supplementary Table 3). A total of 173 compounds increased the *Stm* bacterial load without affecting host cell viability. Four of the hit compounds that decreased the bacterial load (trimethoprim, haloperidol, mibefradil, and ofloxacin) were superior to H-89 in inhibiting intracellular *Stm* (Table 1). Mibefradil again exceeded the inhibitory effect of H-89 in a rescreen (Fig. 3b), while all four compounds consistently and strongly decreased the *Stm* bacterial load. While Fig. 3a and the left panel of Fig. 3b show *z*-score

values, the percentage inhibition of intracellular *Stm* growth is shown in the middle panel of Fig. 3b, expressed as the % of control value, demonstrating the strong inhibitory effect of these HDT compounds on intracellular *Stm*. We next excluded any direct microbicidal activity of these HDT compounds against extracellular *Stm* (Fig. 3c). By contrast, trimethoprim and ofloxacin (both known antibiotics), which were part of the LOPAC library and therefore tested here as well, directly inhibited extracellular *Stm* as expected. The fact that these known antibiotics for *Stm* were hits in our screen further confirms the strength and validity of our approach, showing that we can clearly distinguish antibiotics from host-directed compounds. Taken together, haloperidol (a known HDT inhibitor[19]) and mibefradil (newly discovered here) were confirmed and identified, respectively, as host-directed inhibitors of *Stm*.

Interestingly, a comparison of the *Mtb* and *Stm* HDT compound screening results revealed a highly limited overlap between hits in the two infection models (Fig. 3d). This observation agrees well with reports that *Mtb* and *Stm* arrest vesicle maturation at different stages[16, 32, 33]. Haloperidol was the only compound that inhibited both *Mtb* and *Stm*.

**Identification of HDT compounds using an in silico model**. We next decided to use the above experimental data obtained in our LOPAC screens, and combine these with bioactivity assay-based data available for all 1260 LOPAC compounds in PubChem, to develop a novel bioinformatics predictive model using machine learning. The model was constructed to predict new chemical compounds with host-directed activity against intracellular *Stm* or *Mtb*, based on target protein profiles identified by machine

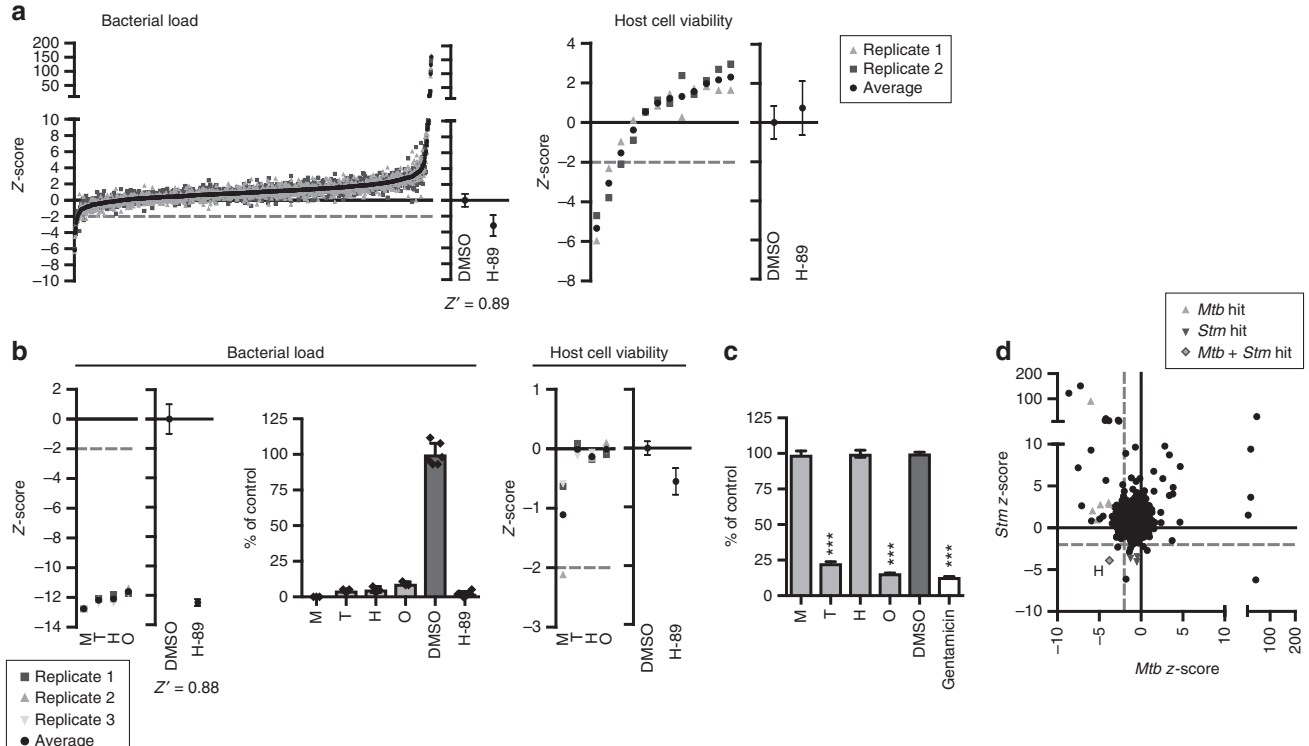

**Fig. 3** Identification of host-directed compounds inhibiting *Stm*. **a** Screen of the LOPAC library in the HeLa-*Stm* infection model using *Stm* constitutively expressing stable DsRed, as in Fig. 2a. **b** Rescreen of the hit compounds from the HeLa-*Stm* screen that were superior to H-89 without affecting cell viability, displayed as in Fig. 2b. The bacterial load is expressed as a z-score in the utmost left panel and as a percentage of control value ± standard deviation in the middle panel to indicate the extent of bacterial inhibition. Compound abbreviations: T = trimethoprim, H = haloperidol, M = mibefradil, O = ofloxacin. **c** Overnight treatment of a *Stm* broth culture with the hit compounds at 10 µM. Gentamicin (50 µg/ml) was used as a positive control. The average bacterial density ± standard deviation of six replicates from a representative experiment out of three experiments is shown. The bacterial load is expressed as a percentage of the DMSO control value to indicate the extent of bacterial inhibition. Statistically significant difference compared to DMSO was tested using a one-way ANOVA ($F_{(5,30)} = 4871$; ***p value < 0.001). **d** Comparison of the *Stm* and *Mtb* primary screening data. Compounds that were superior to H-89 and subsequently confirmed in a rescreen are indicated in gray triangles. H = haloperidol

learning from our own LOPAC screening data (Fig. 4a). In the Supplementary Information, an extended description of the machine-learning methods is provided describing the predictive model. In brief, we first linked all LOPAC compounds to PubChem, and retrieved bioassay data by using a preprocessing pipeline (Supplementary Figure 5a), which identified 1058 confirmed human protein targets for these 1260 compounds. This resulted in a data table comprising all LOPAC compounds annotated with their corresponding impact on intracellular bacterial survival and host cell viability from our screens, expressed as z-scores, and combined with their PubChem bioassay activity for each confirmed human target. An example of the table structure is shown in Supplementary Table 4. This was then used as a training set to learn ensembles of predictive clustering trees (PCTs; Supplementary Figure 5c) to predict the impact on intracellular bacterial survival and host cell viability. We next employed this in silico tool (the learned model) to identify and select candidate compounds from PubChem with predicted host-directed antimicrobial activity. Querying Pub-Chem for compounds that are known to target one or more of the above 1058 confirmed human protein targets yielded 460,580 compounds, which were then annotated with their bioassay data and fed into the predictive model as a testing set. Using the ensembles of PCTs learned from the training data to predict the intracellular bacterial survival and host cell viability z-scores of these 460,580 compounds, we identified 47 candidate compounds predicted to affect intracellular *Mtb* load

(Supplementary Table 5) and 30 compounds predicted to affect intracellular *Stm* load (Supplementary Table 6). From these two lists of compounds, commercially available compounds (Table 2) were ordered and screened in the MelJuSo-*Mtb* and HeLa-*Stm* infection models. As the PubChem BioAssay data contain compound–target relations based only on IC$_{50}$ and EC$_{50}$ values as well as binding constants, the predictive model was able to identify only compound–target interactions rather than the direction of the target effects. Thus, as we were therefore unable to predict whether compounds would actually inhibit or activate their associated targets, predicted negative z-scores might result in experimentally positive z-scores in in vitro intracellular bacterial inhibition tests and vice versa. In the *Mtb* screen, six out of nine compounds predicted to affect the bacterial load indeed decreased or increased the bacterial load (Fig. 4b, left panel). A rescreen of the hit compounds confirmed five out of six hits (VEGFR KI I, ENMD-2076, dovitinib, AT9283, and DAPH 2; Fig. 4b, middle and right panels). The results are shown as z-scores as well as the percentage inhibition of *Mtb* growth expressed as the % of control value, to confirm the strong inhibitory effect of these HDT compounds on intracellular *Mtb* (Fig. 4b, outer right panel).

As compound autofluorescence might result in false-positive z-scores in our assay, we further validated all the confirmed hit compounds independently in classical CFU assays, both in cell lines and in primary human macrophages. The compounds AT9283, ENMD-2076, and dovitinib significantly decreased *Mtb*

CFUs in both MelJuSo cells and human primary Mφs (Fig. 4c; results are shown as percentage inhibition of *Mtb* growth expressed as % of control value). Importantly, AT9283, ENMD-2076, and dovitinib also reduced CFUs in human primary macrophages infected with two different MDR-*Mtb* strains (Beijing family China 16319 and Dutch outbreak 2003-1128; Fig. 4d). These data independently confirm and validate the results obtained in our novel screening and prediction pipeline, and importantly extend the newly identified HDT compounds' effects to intracellular multidrug-resistant bacteria. Finally, none of the compounds directly affected extracellular bacterial growth in liquid cultures, while classical antibiotics (rifampicin) did, confirming that the mode of action of the new HDT compounds is via modulation of host and not direct bacterial mechanisms (Fig. 4e).

Using this same screening and validation approach for *Stm* in the HeLa-*Stm* infection model, we confirmed that two out of four compounds predicted to affect *Stm* survival indeed decreased the bacterial load of *Stm*-infected cells in a primary screen (Fig. 5a, left panel). Both of these hits (opipramol and nafoxidine) were subsequently confirmed in a rescreen (Fig. 5a, middle and right panels; results shown as z-scores and as % inhibition of *Stm* growth expressed as the % of control value). Both hit compounds also reduced the *Stm* bacterial load independently in classical CFU assays (Fig. 5b), again without directly affecting bacterial growth in a liquid overnight *Stm* culture (Fig. 5c), confirming their HDT mode of action. These data therefore confirm and validate our novel screening and prediction pipeline not only for *Mtb*, but also for *Stm*.

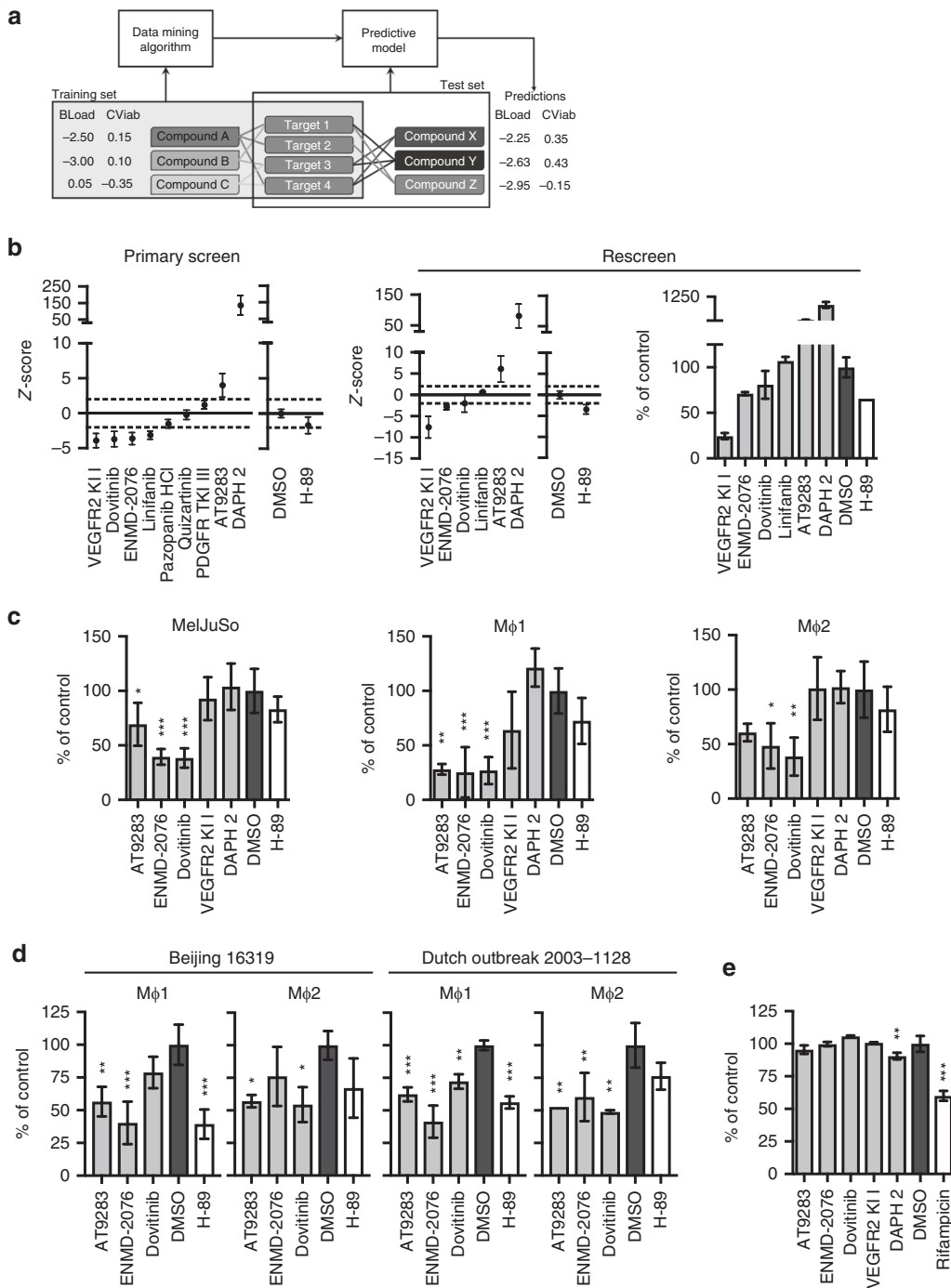

Thus, we have successfully developed and used a data-driven novel in silico predictive model to identify host-directed compounds with antimicrobial activity against intracellular bacteria. The model significantly enhanced the identification of de novo hit compounds (5 out of 9 (55.6%) and 2 out of 4 (50%) for *Mtb* and *Stm*, respectively) compared to random LOPAC library primary screening (126 out of 1260 (10%) for *Mtb* and 185 out of 1260 (14.6%) for *Stm*). In addition, the results were replicated and validated in primary human Mφs infected with *Mtb*, strongly agreeing with and further validating the MelJuSo-*Mtb* model used in our novel flow cytometry-based screening assay.

**RTK signaling is a novel host pathway controlling *Mtb*.** As AT9283, ENMD-2076, and dovitinib are RTK inhibitors[47–49], we used a chemical genetics approach to confirm a role for RTK signaling in host-mediated *Mtb* control. We first retrieved human protein targets of AT9283, ENMD-2076, and dovitinib from the Compound Bioactivity section in ChEMBL (http://www.www.ebi.ac.uk/chembl/) and further downselected targets for which the compounds were annotated as "Active." Because no targets annotated as "Active" could be retrieved for AT9283 and ENMD-2076, we first constructed a STRING protein network and performed gene ontology (GO) analysis using the targets of dovitinib ($n = 86$ proteins; Fig. 6a). Due to the hierarchical organization of GO terms, general cellular and molecular functions tend to be highly enriched in GO term enrichment analyses. Therefore, we focused on identifying the highest-ranked GO terms that described distinct pathways rather than the overall highest-ranked GO terms. As expected from the reported target specificities of dovitinib[49], "transmembrane receptor protein tyrosine kinase signaling pathway" (GO:0007169, false-discovery rate (FDR) 3.82E−33) was the highest-ranking enriched pathway and 40 protein targets participated in this pathway (Fig. 6a and Supplementary Figure 6a). We next verified that both AT9283 and ENMD-2076 target RTKs by retrieving human protein targets from the Target Summary section in ChEMBL and performed an identical STRING analysis (Supplementary Figure 6b, c). Even though this analysis resulted in small networks due to the limited number of studied targets and the lists of targets from the Target Summary section also include nonconfirmed targets, the GO term "transmembrane receptor protein tyrosine kinase signaling pathway" (GO:0007169) was again highly enriched in the

target networks of AT9283 (FDR 1.11E−12) and ENMD-2076 (FDR 6.47E−5).

To independently confirm RTK signaling as a functional pathway that controls intracellular survival of *Mtb*, we next performed an unbiased siRNA screen of the human kinome in the MelJuSo-*Mtb* infection model (Fig. 6b), agnostic to the above data. The siRNA screen identified 20 targets that decreased and 21 that increased the *Mtb* bacterial load while not affecting host cell viability (Table 3). These 41 hit kinases were then used in a STRING protein network and GO analysis. Independently confirming the STRING analysis of the targets of dovitinib, AT9283, and ENMD-2076, also, in this analysis, again "transmembrane receptor protein tyrosine kinase signaling pathway" (GO:0007169, FDR 1.32E−13) was the highest-ranking enriched pathway, and 18 hit kinases from the siRNA screen participated in this pathway (Fig. 6c and Supplementary Figure 6d). Three of the kinases (ABL1, BLK, and NTRK1) were both hits in the siRNA screen and confirmed targets of dovitinib (Fig. 6d). Of these three kinases, only ABL1 was present in the potential target networks of AT9283 and ENMD-2076 (Supplementary Figure 6b, c). However, a lower dissociation constant ($K_i$) is reported in ChEMBL for the interaction between dovitinib and BLK ($K_i$: 12.59 nM) than between dovitinib and ABL1 ($K_i$: 100 nM), suggesting that BLK is targeted more strongly by dovitinib. To identify the top-enriched RTK signaling pathway targeted by dovitinib and siRNA, we used the kinases shown in Fig. 6d in a STRING analysis. This identified the neurotrophin signaling pathway as the top-enriched KEGG pathway (Fig. 6e)[50]. Silencing of neurotrophic receptor tyrosine kinase 1 (NTRK1) resulted in an increased *Mtb* bacterial load (Table 3), establishing a functional link between neurotrophin signaling and *Mtb* survival.

Thus, using independent chemical, genetic, functional, and computational approaches, we find and validate that (1) RTK signaling is a novel host pathway that controls intracellular (MDR)-*Mtb* survival and that (2) repurposable drugs such as dovitinib, AT9283, and ENMD-2076 that target RTK signaling are new candidates for HDT in treating TB, including multidrug-resistant *Mtb*.

**Discussion**

Employing chemical genetic screens complemented with newly developed computational approaches, we have identified host-directed therapy (HDT) compounds and drugs (dovitinib,

---

**Fig. 4** Screen of in silico-predicted chemical compounds active against intracellular *Mtb*. **a** Schematic overview of the in silico predictive model. BLoad = bacterial load z-score, CViab = cell viability z-score. **b** Chemical compound primary screen (left panel) and rescreen (middle panel) at 10 μM in the MelJuSo-*Mtb* infection model using *Mtb* constitutively expressing stable DsRed, expressed as mean z-scores ± standard deviation. Horizontal dashed lines indicate a hit cutoff at a z-score of 2 or −2. The average z-score and standard deviation of the controls (DMSO and H-89) are displayed separately. To indicate the extent of bacterial inhibition, rescreen results are expressed both as z-score and as a percentage of control value ± standard deviation in the right panel. **c** CFU assay of the MelJuSo (left panel) and human primary Mφ1 (middle panel) and Mφ2 (right panel) *Mtb* infection models treated with the validated hit compounds from **b** at 10 μM. Mφ1 and Mφ2 models have been described by Verreck et al.[78] Representative data out of three independent experiments (MelJuSo) and data from a representative donor (Mφs) out of two (Mφ1) or five (Mφ2) different healthy blood bank donors are shown. To indicate the extent of bacterial inhibition, the results are expressed as a percentage of control value ± standard deviation. Number of replicates in the MelJuSo model: AT9283 and ENMD-2076: $n = 6$; dovitinib, VEGFR KI I, and DAPH 2: $n = 5$; DMSO and H-89: $n = 9$. Number of replicates in the Mφ models: AT9283, ENMD-2076, dovitinib, VEGFR KI I, and DAPH 2: $n = 3$; DMSO and H-89: $n = 5$. Statistically significant difference compared to DMSO was tested using a one-way ANOVA (MelJuSo: $F_{(6,39)} = 16.35$; Mφ1: $F_{(6,18)} = 10.88$; Mφ2: $F_{(6,18)} = 5.23$; *p value < 0.05, **p value < 0.01, ***p value < 0.001). **d** CFU assay of the human primary Mφ1 and Mφ2 models infected with two different MDR-*Mtb* strains (Beijing family China 16319 and Dutch outbreak 2003-1128) and treated with the validated hit compounds from **c** at 10 μM. Data ($n = 3$ technical replicates) from a representative donor out of four different healthy blood bank donors, displayed as a percentage of the DMSO control ± standard deviation are shown. Statistically significant differences compared to DMSO were tested using a one-way ANOVA (Mφ1 Beijing: $F_{(4,10)} = 11.43$; Mφ2 Beijing: $F_{(4,10)} = 3.72$; Mφ1 Dutch outbreak: $F_{(4,10)} = 29.09$; Mφ2 Dutch outbreak: $F_{(4,10)} = 8.81$; *p value < 0.05, **p value < 0.01, ***p value < 0.001). **e** Six-day treatment of *Mtb* broth cultures with the hit compounds at 10 μM. The *Mtb* antibiotic rifampicin (20 μg/ml) was used as a positive control. The average bacterial density ± standard deviation of three replicates is shown, expressed as a percentage of the DMSO control. Representative results out of three individual experiments are displayed. Statistically significant difference compared to DMSO was tested using a one-way ANOVA ($F_{(6,25)} = 101.4$; **p value < 0.01, ***p value < 0.001)

**Table 2 Compounds selected from the predictive model output**

| PubChem ID | Compound name | Predicted bacterial load z-score | Predicted cell viability z-score | Reliability score | Primary screen z-score | Rescreen z-score | Activity |
|---|---|---|---|---|---|---|---|
| *Mycobacterium tuberculosis* | | | | | | | |
| **10113978** | Pazopanib·HCl | −2.30 | −0.91 | 0.53 | −1.50 | ND | Receptor tyrosine kinase (RTK) inhibitor |
| **11496629** | AT9283 | −2.27 | −0.95 | 0.54 | **4.01** | **6.09** | JAK/Aurora kinase inhibitor |
| **16041424** | ENMD-2076 | −2.15 | −0.89 | 0.54 | **−3.62** | **−2.83** | RTK/Aurora A inhibitor |
| **11485656** | Linifanib (ABT-869) | −2.11 | −0.87 | 0.54 | **−3.13** | 0.66 | VEGFR/PDGFR inhibitor |
| **10907042** | PDGFR Tyrosine Kinase Inhibitor III | −2.24 | −0.88 | 0.53 | 1.19 | ND | PDGFR inhibitor |
| **9977819** | Dovitinib (TKI-258, CHIR-258) | −2.14 | −0.93 | 0.53 | **−3.70** | **−2.02** | RTK inhibitor |
| **6419834** | VEGFR2 Kinase Inhibitor I | −2.14 | −0.93 | 0.53 | **−3.91** | **−7.63** | VEGFR2 inhibitor |
| **6711154** | DAPH 2 | −2.08 | −0.93 | 0.68 | **136.21** | **80.77** | PKC inhibitor |
| **24889392** | Quizartinib | −2.00 | −0.87 | 0.65 | −0.24 | ND | FLT3 inhibitor |
| *Salmonella typhimurium* | | | | | | | |
| **4416** | Nafoxidine hydrochloride | −1.52 | 0.44 | 0.73 | **−3.06** | **−2.33** | Estrogen receptor modulator |
| **7333** | 1,3-Di-o-tolylguanidine | −1.56 | 0.04 | 0.66 | −0.42 | ND | Sigma 1 receptor agonist |
| **47641** | Naftifine hydrochloride | −1.52 | 0.44 | 0.73 | −0.61 | ND | Fungal squalene epoxidase inhibitor |
| **9417** | Opipramol | −1.51 | 0.15 | 0.69 | **−4.01** | **−4.21** | Sigma receptor agonist |

*ND* not determined
*Z*-scores exceeding the cutoff (2 < *z*-score < −2) are displayed in bold

AT9283, and ENMD-2076) that target human RTK signaling to control intracellular *Mtb* survival, including MDR-*Mtb*. Perhaps, more importantly, our findings pave the way toward identifying additional compounds targeting human RTK signaling to improve control of intracellular *Mtb* infection since all compounds were confirmed to be effective in primary human macrophage infection models.

Current efforts to develop HDT are a topic of interest for infectious diseases and cancer (reviewed recently[38]). In order to be able to screen larger HDT compound libraries for novel leads with activity against intracellular *Mtb* and *Stm*, we have developed a new robust and rapid fluorescence-based intracellular screening assay. This assay allowed us to identify host-directed *Mtb*-inhibiting compounds (SU 6656, quinacrine, SB 216763, GW5074, and tyrphostin AG 494) and host-directed *Stm*-inhibiting compounds (mibefradil), which performed significantly better than our best reference compound H-89, in a LOPAC library drug-repurposing screening effort. We were also able to confirm the activity of previously published HDT compounds in our screening approach (imatinib, D4476, LY-364947, and haloperidol), lending strong plausibility and validity to our strategy.

We next developed a novel in silico model which was data driven and based on known and confirmed targets from public databases, by which we could successfully predict and verify additional compounds with host-directed activity against *Mtb* (dovitinib, AT9283, and ENMD-2076) and *Stm* (nafoxidine and opipramol). Using STRING network analysis, we uncovered RTK signaling as a novel host pathway controlling *Mtb* intracellular survival, which is targeted by compounds identified in this study. Finally, we performed an independent unbiased siRNA screen of the human kinome, which confirmed a role for RTK signaling in control of intracellular *Mtb* survival. Collectively, our results uncover new host-signaling pathways as well as the corresponding active chemical compounds targeting these to control intracellular bacterial infections, including MDR-TB and *Stm*.

Our LOPAC screen provides an important and general proof of principle for drug repurposing, since we successfully identified several candidate compounds that displayed host-directed

antimicrobial activity while their known targets have not previously been associated with infectious diseases. Strikingly, four of the five hit compounds that consistently outperformed H-89 in controlling *Mtb* infection are known to affect (growth factor) RTK signaling. Tyrphostin AG 494, SU 6656, SB 216763, and GW5074 are inhibitors of EGFR, SRC family kinases (SFKs), GSK-3, and RAF1, respectively, which are all kinases participating in RTK pathways[51–55]. In addition to compounds affecting RTK signaling, we identified three other host-directed *Mtb*-inhibiting compounds with vastly different target specificities. First, quinacrine was originally developed as an antimalarial drug but has displayed activity in a myriad of diseases via a wide range of targets[56]. Interestingly, reported targets of quinacrine include AKT1 and NF-κB as well as phospholipase A2[57]. The latter is a central enzyme in the eicosanoid pathway, which was recently shown to be involved in *Mtb* control by balancing the type I interferon response[26]. Second, haloperidol is an antipsychotic drug targeting dopamine receptors[58]. Importantly, haloperidol was recently shown to affect survival of intracellular mycobacteria in a host-directed fashion[19], providing important additional and independent validation of our screening strategy and models. Finally, 3′,4′-dichlorobenzamil is an amiloride-analog Na$^+$/Ca$^{2+}$ exchanger inhibitor[59]. This compound may act by inhibiting Ca$^{2+}$ transport in the cell, as activation of calcineurin by increased Ca$^{2+}$ levels has previously been proposed as a mechanism for inhibition of phagosome maturation in *Mtb*-infected cells[60].

A similar LOPAC library screen in the HeLa-*Stm* infection model resulted in four compounds that more strongly reduced the bacterial load than our reference compound H-89, and mibefradil was further confirmed to surpass H-89's activity in a rescreen. However, H-89 is already a highly potent host-directed inhibitor of *Stm* and all four compounds consistently and significantly reduced the *Stm* bacterial load. Two of the four hit compounds from the primary screen were known antibiotics (trimethoprim and ofloxacin) but these were tested nevertheless in our screen because they were part of the LOPAC. Of the remaining two HDT compounds, haloperidol, which was already found in a previous HDT screen study in TB, was confirmed as a

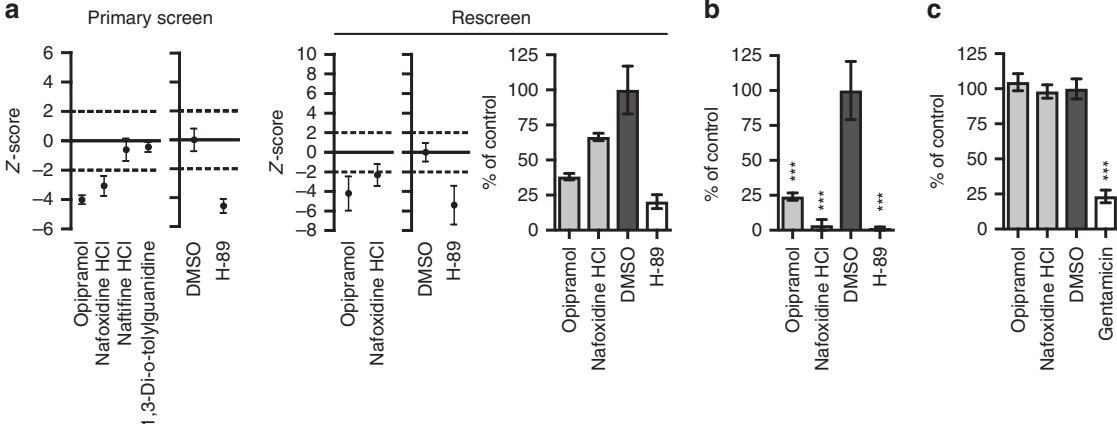

**Fig. 5** Screen of in silico-predicted chemical compounds active against intracellular *Stm*. **a** Chemical compound primary screen (left panel) and rescreen (middle panel) at 10 μM in the HeLa-*Stm* infection model using *Stm* constitutively expressing stable DsRed, expressed as mean *z*-scores ± standard deviation. Horizontal dashed lines indicate a hit cutoff at a *z*-score of 2 or −2. The average *z*-scores and standard deviations of the controls (DMSO and H-89) are displayed separately. To indicate the extent of bacterial inhibition, rescreen results are expressed as a percentage of control value ± standard deviation in the right panel. **b** CFU assay of the HeLa-*Stm* infection model treated with the validated hit compounds from **a** at 10 μM. Representative data out of three independent experiments, displayed as a percentage of the DMSO control are shown. The average ± standard deviation of three replicates is shown. Statistically significant differences compared to DMSO were tested using a one-way ANOVA ($F_{(3,8)} = 56.31$; ***$p$ value < 0.001). **c** Overnight treatment of *Stm* broth cultures with the hit compounds at 10 μM. The *Stm* antibiotic gentamicin (50 μg/ml) was used as a positive control. The average bacterial density ± standard deviation of three replicates is shown, expressed as a percentage of the DMSO control. Representative results out of three individual experiments are displayed. Statistically significant differences compared to DMSO were tested using a one-way ANOVA ($F_{(3,38)} = 579.5$; ***$p$ value < 0.001)

HDT compound with activity against *Mtb*, but we extend these results here to *Stm* as well. These data again show the validity of our screening and prediction approach since we are able to consistently and faithfully confirm already-available knowledge. The data on haloperidol also suggest that this compound may be applicable for HDT in a broader spectrum of intracellular bacterial infections. The second hit compound, mibefradil, is a $Ca^{2+}$ channel blocker[61]. The majority of screening hits in the HeLa-*Stm* infection model exacerbated bacterial loads and even though these compounds can therefore not be used for drug repurposing, all of the identified compounds may be important starting points for gaining deeper mechanistic insight into *Stm*–host interactions. The limited overlap between the hit compounds from *Mtb* and *Stm* screens likely reflects the vastly different intracellular "lifestyles" of these pathogens. Notwithstanding this, several compounds display consistent intracellular antimicrobial activity in both *Mtb* and *Stm* infection models, such as haloperidol. These compounds are therefore promising candidate drugs with wider application against (antibiotic resistant) intracellular bacterial infections.

Selecting hits for follow-up analysis in large (chemical) screens poses substantial challenges. Here, we employed two complementary strategies for screening follow-up. First, as we aimed to identify compounds with superior host-directed antimicrobial activity, we focused on compounds performing better than the reference compound H-89, resulting in a strictly data-driven hit cutoff. Using this strategy, we identified SU 6656, quinacrine, SB 216763, GW5074, and tyrphostin AG 494 as the most promising candidate compounds for TB and mibefradil for salmonellosis, as well as confirmed haloperidol as an attractive drug for HDT against both *Mtb* and *Stm*. Second, as screening outcome may be strongly influenced by compound properties such as solubility, hydrophobicity, concentration, $IC_{50}$, and target selectivity, using a strict cutoff may mask valuable data hidden in the large data set and will be lost to follow-up. We therefore used a complementary follow-up approach by developing an innovative in silico compound predictive model to uncover relevant chemical compound

classes and target profiles in screening data. Focusing on confirmed target profiles by automated extraction of bioassay data from PubChem, we were able to both discern compound targets and predict novel active compounds. As the target profiles were ranked without using a hit cutoff, this approach enabled unbiased validation and follow-up of the primary chemical compound screen. The use of simple numerical values as predictive parameters renders this prediction model highly adaptable and easily applicable to other chemical screens. The model significantly enhanced the identification of de novo hit compounds (55.6% and 50% for *Mtb* and *Stm*, respectively) compared to random LOPAC library primary screening (10% for *Mtb* and 14.6% for *Stm*). Remarkably, the predicted *Mtb* hit compounds AT9283, ENMD-2076, and dovitinib were all (growth factor) RTK inhibitors[47–49].

As inhibitors of RTK signaling molecules were already observed to be overrepresented in the hits from our drug-repurposing screen, our predictive model successfully provided an unbiased validation of this observation and prompted us to further focus our screening endeavor on RTK inhibitors. RTK inhibitors are widely studied in cancer research for their anti-neoplastic properties[62]. Phase II clinical trials have been performed with both AT9283 and ENMD-2076 and dovitinib has already passed phase III clinical trials[63–68] (http://www.clinicaltrials.gov), enabling swift future drug repurposing as host-directed antimicrobials. Our unbiased siRNA screen of the human kinome independently identified and validated RTK signaling as a host pathway regulating *Mtb* survival, identifying BLK, ABL1, and NTRK1 as host kinases, and controlling intracellular *Mtb* and possible drugable targets. BLK is an SFK involved in B cell receptor signaling and the insulin response to glucose uptake in pancreatic islet cells[69, 70]. The non-receptor tyrosine kinase ABL1 was previously linked to mycobacterial infection and its commonly used inhibitor imatinib was shown to exert host-directed Mtb-inhibiting activity in vivo[15, 21], providing independent validation of our siRNA screening. Finally, NTRK1 is an RTK involved in peripheral nervous system development and

synaptic function and plasticity[71]. Various cells of the hemato-poietic lineage have been shown to produce the NTRK ligand nerve growth factor during inflammation and autoimmunity[72] and expression of NTRKs in monocytes has been previously reported[73]. Next to the confirmation of these compound targets by genetic silencing as described here, there were other siRNA hits involved in RTK signaling which might represent as-yet-

unknown molecular targets for these or other hit compounds. Conversely, confirmed compound targets that were not identified in our siRNA screen may still contribute to *Mtb* control due to redundancy and possible incomplete genetic knockdown inherent to siRNA screens.

A role for growth factors in mycobacterial infection has been previously reported. The growth factor VEGF was linked to

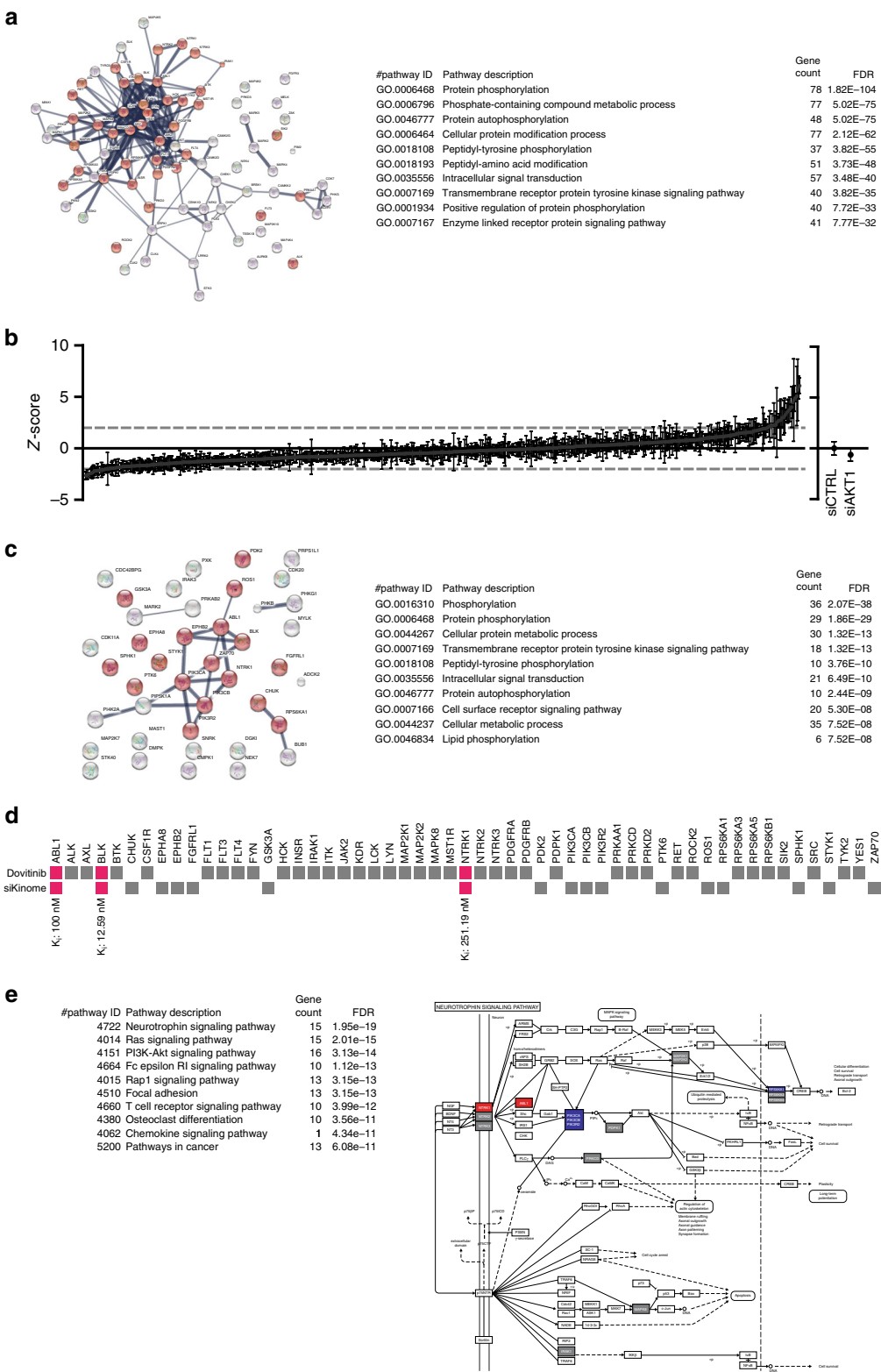

mycobacterial infection in a zebrafish *Mycobacterium marinum* (*Mmar*) infection study[28] as well as in a rabbit *Mtb* infection model[27]. However, in both studies, the reported effect of VEGF

**Table 3 siKinome screen hits in the MelJuSo-*Mtb* infection model**

| GenBank accession | Gene symbol | *z*-score |
|---|---|---|
| NM_006213 | PHKG1 | −2.65 |
| NM_133494 | NEK7 | −2.65 |
| NM_018425 | PI4KII | −2.50 |
| NM_032017 | MGC4796 | −2.47 |
| NM_019884 | GSK3A | −2.44 |
| NM_014975 | SAST | −2.42 |
| NM_006219 | PIK3CB | −2.34 |
| NM_001079 | ZAP70 | −2.25 |
| NM_005157 | ABL1 | −2.19 |
| NM_012119 | CCRK | −2.18 |
| NM_017525 | HSMDPKIN | −2.17 |
| NM_007199 | IRAK3 | −2.16 |
| NM_001715 | BLK | −2.15 |
| NM_001278 | CHUK | −2.14 |
| NM_000293 | PHKB | −2.11 |
| NM_002611 | PDK2 | −2.09 |
| NM_017771 | PXK | −2.08 |
| NM_005399 | PRKAB2 | −2.02 |
| NM_021923 | FGFRL1 | −2.01 |
| NM_004717 | DGKI | −2.00 |
| NM_005027 | PIK3R2 | 2.02 |
| NM_021972 | SPHK1 | 2.12 |
| NM_001100594 | SNRK | 2.20 |
| NM_006218 | PIK3CA | 2.26 |
| NM_175886 | PRPS1L1 | 2.26 |
| NM_005975 | PTK6 | 2.27 |
| NM_033487 | CDC2L1 | 2.51 |
| NM_001007792 | NTRK1 | 2.62 |
| NM_001081562 | DMPK | 2.69 |
| NM_001006665 | RPS6KA1 | 2.72 |
| NM_016308 | UMP-CMPK | 2.77 |
| NM_002944 | ROS1 | 3.31 |
| NM_004336 | BUB1 | 3.50 |
| NM_182493 | MLCK | 3.58 |
| NM_001006943 | EPHA8 | 3.70 |
| NM_052853 | ADCK2 | 3.80 |
| NM_018423 | STYK1 | 4.02 |
| NM_001039468 | MARK2 | 4.46 |
| NM_145185 | MAP2K7 | 4.85 |
| NM_003557 | PIP5K1A | 6.10 |
| NM_017449 | EPHB2 | 7.68 |

was primarily systemic rather than (sub)cellular, inducing enhanced angiogenesis in granulomas. Our data strongly suggest that an intracellular response to growth factor receptor signaling via RTKs may be another important determinant for mycobacterial infection outcome. Interestingly, Oehlers et al. used pazopanib, one of the compounds identified by our predictive model to show an inhibitory effect of VEGF receptor (VEGFR) inhibition on vascularization around nascent granulomas in their model. Though not meeting our strict hit selection criteria, pazopanib statistically significantly (*z*-score −1.50) decreased *Mtb* loads in our screen (and thus in the absence of a vascular system), suggesting that cellular *Mtb* inhibition by pazopanib might precede or complement the vascularization effects observed in vivo by Oehlers et al. Additionally, epidermal growth factor receptor (EGFR) signaling has previously been linked to mycobacterial infection through a chemical screen identifying EGFR inhibitor gefitinib as a compound that restricts *Mtb* growth[31]. Our study significantly expands this knowledge by introducing additional RTK-targeting compounds that can be used for drug repurposing, including compounds targeting VEGFR (dovitinib) and EGFR (tyrphostin AG 494) signaling.

Our in silico predictive model successfully identified two compounds (nafoxidine, an estrogen receptor modulator and opipramol, a sigma receptor agonist) with host-directed *Stm*-inhibiting activity. Interestingly, haloperidol (a hit in both the *Mtb* and *Stm* LOPAC screens) was previously reported to interact with sigma receptors with high affinity[74], suggesting mechanistic involvement of sigma receptors in host control of intracellular bacteria.

Finally, the development and validation of a novel fluorescence-based screening assay that is able to rapidly quantify intracellular bacterial infection in human cells, as described here, we think that it is important since it helps shortening the readout from a classical 3-week CFU assay for *Mtb* to 24–72 h using flow cytometry. The assay is highly reproducible, medium throughput, provides an excellent assay window, and is suitable for screening both chemical compound and siRNA libraries. Taking advantage of the previously reported phagocytic capability of melanocytes[41], we also report the human melanoma cell line MelJuSo as a novel model for *Mtb* infection studies, particularly for studies encompassing chemical and RNAi screens. Importantly, our assay faithfully reproduces the inhibitory effect of several previously published host-directed compounds (that were identified using different strategies, approaches, and models) on *Mtb* intracellular survival, as well as HDT results obtained in human primary macrophages, further validating our novel infection model.

**Fig. 6** Identification of host kinases controlling intracellular *Mtb* survival. **a** STRING network of confirmed targets of dovitinib retrieved from the ChEMBL repository Compound Bioactivity section (left panel). Individual proteins are displayed as nodes. Lines represent protein–protein interactions and the thickness of the lines indicates confidence. Proteins participating in the "transmembrane receptor tyrosine kinase signaling pathway" are displayed in red. The top 10 enriched GO terms in the "Biological Function" category are displayed along with the number of genes/proteins annotated with the indicated GO terms and the false-discovery rate (FDR) of the enrichment (right panel). **b** Results of a siRNA screen of the human kinome in the MelJuSo-*Mtb* infection model using *Mtb* constitutively expressing destabilized DsRed, expressed as *z*-scores. The average *z*-score ± standard deviation for each siRNA pool is displayed. A hit cutoff at $z = 2$ or $z = -2$ is displayed as a dashed line. The average *z*-score and standard deviation of the controls (siCTRL and siAKT) are displayed separately. SiCTRL: nontargeting siRNA pool. **c** STRING network of the siRNA screen hits (left panel) is displayed along the top 10 enriched GO terms in the "Biological Function" category (right panel), as in **a**. **d** Participation of individual targets of dovitinib (top row) or hits from the siRNA screen (bottom row) in the "transmembrane receptor tyrosine kinase signaling pathway" are indicated by filled squares. Proteins that are both targeted by dovitinib and were a hit in the siRNA screen are shown in magenta. Dissociation constants ($K_i$) retrieved from ChEMBL are shown below for the interaction between dovitinib and ABL1, BLK, and NTRK1. **e** STRING analysis to identify enriched KEGG pathways using the kinases from **d**. The top 10 enriched KEGG pathways along with the number of genes/proteins annotated with the indicated GO terms and the FDR of the enrichment are displayed (left panel). Involvement of individual proteins is overlaid on the "neurotrophin signaling pathway" KEGG pathway retrieved from the Kyoto Encyclopedia of Genes and Genomes (http://www.genome.jp/kegg/). Proteins in gray are targeted by dovitinib only, blue proteins were siRNA screening hits, and proteins in red are both targeted by dovitinib and silencing of these genes affected the *Mtb* bacterial load

In conclusion, the results from our chemical, genetic, and novel bioinformatics approach provide an important proof of concept of HDT for intracellular infections, such as (MDR) TB and salmonellosis. Moreover, our results identify human RTK signaling as a signaling pathway targetable by novel repurposable drugs, providing a new and promising therapeutic starting point for drug development against *Mtb*, including MDR-*Mtb*.

## Methods

**Reagents**. H-89 dihydrochloride, D4476, IC261, LY-364947, DAPH 2, nafoxidine hydrochloride, 1,3-di-o-tolylguanidine, naftifine hydrochloride, opipramol, rifampicin, kanamycin, and the library of pharmacologically active compounds (LOPAC) were purchased from Sigma-Aldrich, Zwijndrecht, the Netherlands. Hygromycin B was acquired from Life Technologies-Invitrogen, Bleiswijk, the Netherlands. Imatinib mesylate was from Enzo Life Sciences, Raamsdonksveer, the Netherlands. VEGFR2 kinase inhibitor I and ampicillin were purchased from Calbiochem Merck-Millipore, Darmstadt, Germany. Pazopanib HCl, AT9283, and linifanib (ABT-869) were acquired from Selleck Chemicals, Munich, Germany. Quizartinib was purchased from MedChemExpress, Stockholm, Sweden. Santa Cruz BioTechnology, Heidelberg, Germany was the supplier of PDGFR tyrosine kinase inhibitor III. Dovitinib (TKI-258, CHIR-258) was from APExBIO, Houston, TX, USA. The siKinome library was acquired from Thermo Fisher Dharmacon, Waltham, MA, USA.

**Cell culture**. HeLa cells and the MelJuSo human melanoma cell line were maintained at 37 °C and 5% $CO_2$ in Gibco Iscove's Modified Dulbecco's Medium (IMDM; Life Technologies-Invitrogen) with 10% fetal bovine serum (FBS, Greiner Bio-One, Alphen a/d Rijn, the Netherlands), 100 units/ml penicillin and 100 μg/ml streptomycin (Life Technologies-Invitrogen). Proinflammatory Mφ1s and anti-inflammatory Mφ2s were generated from monocytes isolated from whole blood of healthy donors by FICOLL separation and CD14 MACS sorting (Miltenyi Biotec, Teterow, Germany) followed by 6 days of differentiation with 5 ng/ml granulocyte macrophage–colony-stimulating factor (GM–CSF; BioSource Life Technologies-Invitrogen) or 50 ng/ml macrophage–colony-stimulating factor (M–CSF; R&D Systems, Abingdon, UK), respectively, as previously reported[75]. Mφs were cultured in Gibco Roswell Park Memorial Institute (RPMI) 1640 medium (Life Technologies-Invitrogen) with 10% FBS and 2 mM L-alanyl-L-glutamine (PAA, Linz, Austria).

**Bacterial culture**. Bacterial strains used are displayed in Supplementary Table 1. Mycobacteria were cultured in Difco Middlebrook 7H9 broth (Becton Dickinson, Breda, the Netherlands) supplemented with 10% ADC (Becton Dickinson), 0.5% Tween-80 (Sigma-Aldrich), and appropriate antibiotics. *Stm* was cultured on Difco Luria-Bertani (LB) agar (Becton Dickinson) or in Difco LB broth (Becton Dickinson) supplemented with appropriate antibiotics.

**Stm and Mtb infections**. One day before infection, mycobacterial cultures were diluted to a density corresponding with early log-phase growth (optical density at 600 nm ($OD_{600}$) of 0.4). *Stm* was grown either in LB broth or on LB agar with appropriate antibiotics. After overnight incubation, *Stm* liquid cultures were diluted 1:33 and cultured for an additional 3–4 h, while plate-grown *Stm* was suspended in PBS by rinsing the agar plates. Bacterial density was determined by measuring the $OD_{600}$ and the bacterial suspension was diluted in cell culture medium without antibiotics to reach a multiplicity of infection (MOI) of 10 (unless indicated otherwise). Accuracy of bacterial density measurements was verified by a standard colony-forming unit (CFU) assay. Cell cultures (HeLa for *Stm* infections and MelJuSo for *Mtb* infections), seeded in 96-well flat-bottom plates as described below, were inoculated with 100 μl of the bacterial suspension, centrifuged for 3 min at 800 rpm, and incubated at 37 °C/5% $CO_2$ for 20 min if infected with *Stm* or 60 min if infected with *Mtb*. The plates were then washed with culture medium containing 30 μg/ml gentamicin sulfate (Lonza BioWhittaker, Basel, Switzerland) and incubated at 37 °C and 5% $CO_2$ in a medium containing 5 μg/ml gentamicin and the indicated chemical compounds until readout by flow cytometry or CFU, as indicated.

**Chemical compound treatment**. A total of 10,000 HeLa or MelJuSo cells were seeded per well in 96-well flat-bottom plates or 300,000 primary macrophages were seeded per well in 24-well plates in appropriate culture medium without antibiotics 1 day prior to infection with *Mtb* or broth-grown *Stm*. Infected cells were treated overnight with chemical compounds at 10 μM (unless indicated otherwise) or DMSO at equal v/v in a medium containing 5 μg/ml gentamicin.

**siRNA transfections**. A total of 3,000 HeLa or MelJuSo cells were reverse-transfected with ON-TARGETplus siRNA pools (Thermo Fisher Dharmacon, Waltham, MA, USA) at a 50 nM concentration using 0.2 μl of Dharmafect1 (Thermo Fisher Dharmacon) per well in a flat-bottom 96-well plate in appropriate culture medium without antibiotics. Knockdown efficiency was verified by immunoblotting at the indicated time points. Cells transfected with siRNA were

infected with *Mtb* at MOI 1000 24 h post transfection and incubated for an additional 48 h and infections with agar-grown *Stm* were carried out at MOI 500 72 h post transfection and incubated overnight, unless indicated otherwise.

**Colony-forming unit assay**. CFU assays were performed using the track dilution method described previously[76]. In short, bacterial suspensions were serially diluted and 10-μl drops were plated on square agar plates, which were subsequently placed at an angle to allow the drops to spread over a larger surface area.

**Bacterial growth assay**. A volume of 100 μl of *Stm* or *Mtb* culture ($OD_{600}$ of 0.1) was plated in a flat-bottom 96-well plate containing 100 μl of indicated chemical compounds at 20 μM in LB (*Stm*) or 7H9 (*Mtb*) broth. The plate was incubated at 37 °C overnight for *Stm* or during a period of 15 days for *Mtb* and absorbance was measured at a 550-nm wavelength on a Mithras LB 940 plate reader (Berthold Technologies, Bad Wildbad, Germany).

**Fluorescence microscopy**. A total of 100,000 HeLa or MelJuSo cells were grown on glass coverslips (Menzel-Gläser, Braunschweig, Germany) in 24-well plates and infected as described above. Samples were fixed for 30 min at RT with 4% paraformaldehyde, embedded in VectaShield with DAPI (Brunschwig Chemie, Amsterdam, the Netherlands), and examined on an Axioskop 2 fluorescence microscope (Carl Zeiss, Sliedrecht, the Netherlands).

**STRING analysis**. Protein interaction networks were generated using STRING version 10 (http://string-db.org/)[77] using experiments and databases as data sources and a minimal confidence score of 0.4.

**Statistics**. Student's *t* test, one-way ANOVA, and linear regression were performed using GraphPad Prism version 6.0 for Mac OS X (GraphPad Software, San Diego, CA, USA; www.graphpad.com).

**Data availability**. The data that support the findings of this study are available from the corresponding authors on request.

**Code availability**. The code of the machine-learning software CLUS that was used to build the in silico models for predicting compound activity is available for download from the SourceForge repository (at https://sourceforge.net/projects/clus/).

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

## Acknowledgements

This project was funded by the European Union's Seventh Programme for research, technological development, and demonstration under grant agreement No. PhagoSys HEALTH-F4-2008-223451; NEWTBVAC HEALTH.F3.2009 241745, TANDEM project Grant Agreement No. 305279, MAESTRA ICT-2013-612944, and HBP SGA1 ICT-2013-604102. We also gratefully acknowledge the support of the Netherlands Organization for Health Research and Development (ZonMw-TOP grant 91214038) and Technology Foundation STW (grant 13259). The funders had no role in study design, data collection and analysis, decision to publish, or preparation of the manuscript. *Stm* strains expressing (destabilized) DsRed constructs were kindly provided by Prof. dr J.J. Neefjes (LUMC, Leiden, the Netherlands) and we are grateful to Dr J. Bestebroer (VUMC, Amsterdam, the Netherlands) for mycobacterial reporter constructs. The LOPAC library was kindly provided by Prof. dr H. Ovaa (LUMC, Leiden, the Netherlands). We thank Dr C. Kuijl for assay development guidance and Dr S.A. Joosten for critically reviewing the manuscript.

## Author contributions

C.K. designed and performed experiments and wrote the manuscript; D.K. developed the in silico predictive model and revised the manuscript; M.T.H. designed and performed experiments and revised the manuscript; E.S. and O.R. performed experiments; K.F. developed fluorescent reporter constructs; L.W. transformed bacteria with fluorescent reporter constructs; N.S. supervised and designed experiments; S.D. supervised development of the in silico predictive model and revised the manuscript; and M.H. and T.O. supervised, designed experiments, revised the manuscript, and had primary responsibility for the final content of the manuscript. All authors have read and approved the final manuscript.

## Additional information

**Competing interests:** The authors declare no competing financial interests.

