## [Peer Review File · Nature Communications]

Reviewers' comments:

Reviewer #1 (Remarks to the Author):

The manuscript by Korbee et. al. attempts to combine a chemical screen with a bioinformatics pipeline, followed by a genetic screen, to identify host-directed drugs for intracellular bacterial infections. While this could potentially have been an interesting study it, unfortunately, suffers from several shortcomings. First, the manuscript is written in an extremely confusing manner, lacking clarity on several key points, and requires many readings before one can come to grips with what the authors have actually attempted. This is particularly true for the section on the in silico prediction algorithm (in reality a pipeline rather than an algorithm). The description of the steps involved lack the requisite detail and the reader is left struggling to understand what was done. The supplementary material is also of no particular help in this regard. Additionally, there are also concerns on both the experimental design and the value of the data generated. These are:

1. The reason for use of a melanoma cell line for Mtb infection in the screen is not clear. While these cells may be capable of phagocytosing Mtb, the question of whether the intracellular environment experienced by the bug is similar to that it would counter within a macrophage has not been clarified. While macrophages would respond to an intracellular infection through an array of anti-bacterial responses, it is unlikely that melanoma cells will be anywhere as aggressive. But surely this would be an important consideration when developing an assay for host-directed drugs? A similar argument can be made for the use of HeLa cells as an infection model for Stm.
2. Continuing on the assays, Suppl Fig 1A shows that only about 5%, or less, of the HeLa cells were infected with Stm under the conditions employed. Similarly, the fraction of melanoma cell infected with Mtb was of the order of about 30%. It is doubtful if such low infection loads would yield sensitive and reliable readouts. Indeed the correlation plots shown in Suppl Fig 1F-E&G are in fact very poor with points just bunching at either end rather than displaying a true linear relationship.
3. The majority of the screen results are given in terms of z-scores and it is difficult to determine how potent the hits really are. In the one instance where the efficacy data is actually provided (Fig. 2D&E) the effect on Mtb infection is only marginal with a reduction of only about 50% or less in CFU values. It is questionable whether such compounds can really be considered as drugs.
4. The use of H89 as an Akt inhibitor is puzzling. At least to this reviewer's knowledge, H89 is a PKA inhibitor. For Akt, more efficient and specific inhibitors (e.g. MK2206) are available.
5. The use of predictive models, a training set and a testing set for the identification of novel hits is claimed to represent a new algorithm. However from all the steps it is clear that this is just a pipeline that links already established methods to identify new hits. Describing this as an algorithm, therefore, is a bit misleading.
6. Further, to validate the above protocol, the authors state "Commercially available compounds predicted to strongly decrease bacterial load were selected..." But this statement is not accompanied anywhere by an explanation of how such a selection was done. Was it on the basis of binding of compounds in PubChem to targets for the hits in the LOPAC screen? Even if that is the case, how do they predict that the compounds will decrease bacterial load, without affecting host cell viability?
7. The description of how the predictive tool was developed is very confusing, with no rationale provided for many of the steps involved. Some pertinent points here are:
 - There is no clarity on how percentage of active compounds was calculated. What values were used?
 - The authors build a training set that is used to generate a predictive model, which then identifies new hits from the testing set data. But there is no explanation for the criteria (e.g. physical and chemical properties) that were employed for the matching of similar compounds.
 - How was the Z-score and reliability score calculated? And how does the Z-score link with the

bacterial load and cell viability?

- Multiple predictive models were constructed using different bootstrap replicates of the training data set, and the average of different bootstrap replicates were applied for overall prediction. But what are these replicates? An adequate explanation of this is wanting.

8. Finally, at least in the view of this reviewer, the manuscript is replete with experimental redundancies. For instance, in developing the predictive tool, the entire LOPAC library was matched with PubChem to identify 1058 protein targets. The rationale for this is puzzling. Why not just match the hits from the LOPAC screen? And at the end of this complicated exercise the authors end up with a very small number of predicted hits: 9 for Mtb and 4 for Stm. This small number is surprising because a simple virtual screen of the 460,580 compounds against a single target protein should have yielded more hits depending on the binding energy cut-off used. The last segment where a human kinome RNAi screen was used to validate the handful of predicted hits is again another case of needless redundancy. Was an entire kinome scan really necessary? And if the only conclusion from this was that some RTKs are important for intracellular bacterial survival, does it add any value given the extensive information already available in the literature?

Reviewer #2 (Remarks to the Author):

Korbee et al. describe a novel and fast FACS based screening assay designed for screening of drug candidates against intracellular bacteria either Salmonella or Mycobacterium tuberculosis. They used this assay in a small 1200 compounds screening experiment and identified several compounds that inhibited these pathogens inside their infection model. At the second stage they used a bioinformatics approach to identify inhibitors of confirmed targets within the human host and tested those using their novel assay using a set of inhibitors and siRNA approach.

Overall this is a very interesting and novel approach yet the paper in its current form suffers from multiple deficiencies described below.

Major deficiencies

1. The hella cells are good model for Salmonella infection assay (salmonella infects epithelial cells) but the melanoma derived cells are poor model for Mtb infection. The data demonstrated the poor performance of this model in the case of Mtb. All the compounds (and not only selected ones) should be tested on monocytes derived or primary phagocytotic cells infected with Mtb and the results should be confirmed in terms of CFU's counts from.
2. The efficacy results are presented in Z' score values. While these can demonstrate that the assay is robust and reproducible it does not provide a comparable or reliable efficacy measure. Intracellular MIC's or (because of the poor activity of the compounds) % inhibition would be a better measure.
3. I could not understand the rationale for expressing the activity against H-89. Are there issues with independent assessment of the standalone activity of each compound? H-89 can be used as a control but the description of the results by means of the efficacy assay should be better explained.
4. As the results used a single concentration of inhibitors each of the compound should be verified using dose and time dependent experiments.
5. The effect of all compounds and SiRNA clones should be controlled to rule out effect on cells metabolic ability to reflect the fluorescent marker (usually done on beads coated with fluorescence protein).
6. A better literature search should have been performed. I found that bromo- and Halo- peridols were identified already as antimycobacterial agents working in alone and in synergy with antibiotics.
7. I found the mix of Salmonella and Mtb experiments confusing and difficult to assess. I think it would be more appropriate to provide a linear description of each one of the organisms and then compare the two.

Minor points.

1. A more appropriate term for your assay is medium throughput.
2. The term high content is used inappropriately. High content usually involves assessment of multiple parameters such as expression of proteins, viability, toxicity, etc. Simple measurement of fluorescence is not high content.
3. The term algorithm is usually associated with flow charts or disclosure of the program source code. From my understanding, the process that has been performed is manual bioinformatics analysis of multiple parameters.
4. A better description of the novelty of the finding should be provided. After all, screening of signaling inhibitors and RNAi against intracellular MTb has been performed in the past, as well as a repurposing approach to test for MTb inhibitors.
5. As this paper describes a medicinal chemistry approach for drug discovery, I encourage the authors to use medicinal chemistry jargon and terms or relate their units (z' score and absorbance).
6. The term druggable should be better explained. As all compounds only slightly affect mycobacterial growth, how could they be considered drugs?

Response to reviewers' comments

Reviewer #1

- 1. The reason for use of a melanoma cell line for *Mtb* infection in the screen is not clear. While these cells may be capable of phagocytosing *Mtb*, the question of whether the intracellular environment experienced by the bug is similar to that it would counter within a macrophage has not been clarified. While macrophages would respond to an intracellular infection through an array of anti-bacterial responses, it is unlikely that melanoma cells will be anywhere as aggressive. But surely this would be an important consideration when developing an assay for host-directed drugs? A similar argument can be made for the use of HeLa cells as an infection model for *Stm*.**

RESPONSE: We understand the reviewer's initial reservation regarding the use of a melanoma cell line as a model for *Mtb* infections. Before explaining the choice for the MelJuSo (and HeLa) cell lines, we would like to emphasize that all our compound hits are validated in primary human macrophages as a key gating criterion. Thus, even though cell lines are used for discovery of candidate target molecules and circuits, only those that validated in primary human macrophages qualify as real hits in the study (as exemplified in Figure 3C and Supplementary Figure 1G).

In search for a suitable human cell line model for *Mtb* infection we initially tested several available cell lines for potential use in medium/high throughput chemical genetic screens. The often-used THP-1 human monocytic cell line, however, has a major disadvantage in that it requires PMA stimulation to induce a macrophage-like, adherent phenotype, which profoundly alters PKC signaling. Screening compounds and siRNA libraries on such a PMA-modulated background would complicate the identification of target proteins, and preclude study of molecules in the PKC pathway. In contrast, the well documented MelJuSo cell line (e.g. Cell 145:268-283) does not require such additional stimuli. Moreover, it is transfectable with siRNA with high efficiency, is highly homogeneous and has phagocytic capacity. We had already shown in the past that human melanocytes are efficient antigen presenting cells (APC) that process and present mycobacteria to CD4 T cells in a HLA class II restricted fashion (J. Immunol.151:7284). We have also worked with MelJuSo as APC in the past to dissect molecular pathways of HLA class II presentation, identifying the role of HLA-DM and HLA-DO (e.g. Current Biology 7:950-957; J ExpMed 191:1127). For these reasons, we decided to develop MelJuSo as a novel human intracellular model for *Mtb* infection, allowing medium/high throughput chemical and genetic screens.

Importantly, we first further validated the relevance of the MelJuSo model by reproducing the effect of host-directed *Mtb*-inhibiting compounds that had already been reported in literature. To better emphasize this key point, we have now moved the figure showing this important bridging data from the Supplementary Information to the main manuscript (new Figure 1). We have also added additional text to the manuscript explaining the rationale behind the use of the MelJuSo cell line and explaining the advantages of this model for chemical genetic screens (page 3 lines 77-88 and page 5 lines 113-128 in the main manuscript and page 1 lines 12-16 in the Supplementary Information). Finally, to highlight that MelJuSo cells are equipped with important antimicrobial host defense mechanisms, we now report the expression of key genes coding for antimicrobial effectors and regulators based on existing microarrays (courtesy

of Prof. Dr. J.J. Neefjes, Leiden University Medical Center, Leiden, The Netherlands; confidential, personal communication). We have attached the raw expression data as an Excel file with this letter. [File Redacted]

- 2. Continuing on the assays, Suppl Fig 1A shows that only about 5%, or less, of the HeLa cells were infected with *Stm* under the conditions employed. Similarly, the fraction of melanoma cell infected with *Mtb* was of the order of about 30%. It is doubtful if such low infection loads would yield sensitive and reliable readouts. Indeed the correlation plots shown in Suppl Fig 1F-E&G are in fact very poor with points just bunching at either end rather than displaying a true linear relationship.**

RESPONSE: We understand the reviewer's concerns regarding the infection rates. Using an identical *Mtb* infection protocol in primary human macrophages, results routinely are in very similar ranges of infection rates as observed in MeJuSo. However, in these human macrophages both *Mtb* and *Stm* infection rates differ between experiments and between donors, ranging from ~5% to ~60%. In other published studies mostly CFU (colony forming units as a measure of the number of bacteria) assays are used, but these provide no indication of the actual percentage of infected cells at all, so comparison of our results with literature is challenging. We have performed extensive *Mtb* and *Stm* MOI titrations in primary macrophages, aiming for the highest possible infection rate without affecting host cell viability, and consistently find that a significant proportion of cells remains uninfected (this is in fact the topic of an independent project).

We agree with the reviewer that Supplementary Figures 1E and 1G were not convincing and that it would have been better to base these on a bacterial titration rather than a limited number of different compound treatments. As these figures were merely a different representation of data readily available in the bar graphs in Supplementary Figure 1, we have therefore removed them from the manuscript.

- 3. The majority of the screen results are given in terms of z-scores and it is difficult to determine how potent the hits really are. In the one instance where the efficacy data is actually provided (Fig. 2D&E) the effect on *Mtb* infection is only marginal with a reduction of only about 50% or less in CFU values.**

RESPONSE: We would like to mention that the use of z-scores is quite common in reporting data from chemical screens (see also reply to point 7c below for further details). In contrast to many other studies on novel compounds targeting intracellular *Mtb* that typically study the effects of compounds over a time window of several days and sometimes add these in multiple successive doses, we instead have decided to test our compounds after only a single overnight treatment. This strategy allows rapid screening and reveals only the most potent compounds. The effects we see on *Mtb* with these compounds have not been optimised further, which will be the focus of further translational medicine studies in our lab, using animal models (zebrafish, mice). Thus our model is extremely stringent and designed to identify only the best hits; to exemplify this, Imatinib did not pass our strictest selection criteria, despite its reported efficacy *in vivo* (see: Napier *et al.* Cell Host and Microbe 10, 475–485). We believe that our strategy helps finding compounds that might display superior activity against *Mtb* and *Stm*.

It is questionable whether such compounds can really be considered as drugs.

RESPONSE: Despite the incompletely sterilizing effects of individual compounds in our single-dose, short term (overnight) setup, we would contend that our thus defined host-directed compounds can be considered novel drugs as they may be used in conjunction with classical antibiotics to 1) kill residual “dormant/nonreplicating” bacteria that are metabolically inactive and thus do not respond to antibiotics; 2) shorten current lengthy TB treatment regimens (6-9 month regimen); and 3) help treating MDR/XDR-TB resistant to most available drugs. Moreover it is well possible that combinatorial regimens of different classes of HDT compounds may exert superior activity against intracellular bacteria by targeting multiple synergizing host pathways. In cancer, combinatorial immunotherapy is becoming more and more in vogue, using e.g. immune checkpoint inhibitors in combination with kinase inhibitors (e.g. Nature 543:728, 2017).

4. **The use of H89 as an Akt inhibitor is puzzling. At least to this reviewer’s knowledge, H89 is a PKA inhibitor. For Akt, more efficient and specific inhibitors (e.g. MK2206) are available.**

RESPONSE: We indeed agree that H89 has mostly been known as a PKA inhibitor. However, in our previous paper by Kuijl, C., *et al.*, (Intracellular bacterial growth is controlled by a kinase network around PKB/AKT1. Nature 450, 725–730; 2007) we have demonstrated that H89 mediated its effects on both *Stm* and *Mtb* inhibition by inhibiting AKT1 rather than PKA. This was confirmed by using additional chemical inhibitors as well as genetically (RNAi). We have added additional text and the above reference to the main manuscript on page 3 lines 60-66 to explain this.

5. **The use of predictive models, a training set and a testing set for the identification of novel hits is claimed to represent a new algorithm. However from all the steps it is clear that this is just a pipeline that links already established methods to identify new hits. Describing this as an algorithm, therefore, is a bit misleading.**

RESPONSE: We agree with this comment, and have updated the text accordingly, describing our new approach as a predictive model rather than an algorithm.

6. **Further, to validate the above protocol, the authors state “Commercially available compounds predicted to strongly decrease bacterial load were selected...” But this statement is not accompanied anywhere by an explanation of how such a selection was done. Was it on the basis of binding of compounds in PubChem to targets for the hits in the LOPAC screen? Even if that is the case, how do they predict that the compounds will decrease bacterial load, without affecting host cell viability?**

RESPONSE: We fully agree with the reviewer that this section was lacking clarity, and thank the reviewer for pointing this out. We have completely rewritten this section, and now provide a much more detailed explanation of how the *in silico* experiments were performed, both in the main manuscript (pages 6-7, lines 159-187) and in the Supplementary Information (pages 3-6, lines 95-197). We have also updated Figure 3A and Supplementary Figure 5A to further clarify the prediction pipeline and *in silico* methods. The extended explanation of the methods should now also clarify the selection criteria used that resulted in the final set of selected compounds. In brief, we linked all LOPAC compounds to PubChem, and bioassay data were retrieved, identifying 1058 confirmed human protein targets of these 1260 compounds. This resulted in a data table of all LOPAC compounds annotated with their bacterial load and cell viability z-scores from the performed screens, as well as their PubChem bioassay activity. This was then used as a

training set to learn ensembles of predictive clustering trees to predict impact on bacterial survival and host cell viability of compounds in the PubChem repository. Querying PubChem for compounds confirmed to target at least one of the 1058 identified proteins yielded 460,580 compounds, which were then annotated with their bioassay data and fed into the predictive model as a test set to predict their bacterial load and cell viability z-scores. Together with the predictions, we calculated a reliability score based on the variance of the predictions in the ensemble. Based on the predicted bacterial load and cell viability z-scores, we generated a list of candidate hit compounds. We then further filtered this list of candidates by only considering predictions that are most reliable (using the reliability score) and finally selected compounds that were commercially available for further experiments.

7. **The description of how the predictive tool was developed is very confusing, with no rationale provided for many of the steps involved. Some pertinent points here are:**

a. **There is no clarity on how percentage of active compounds was calculated. What values were used?**

RESPONSE: We agree again with the reviewer that this description was lacking clarity, and thank the reviewer for pointing this out. In answer to point a, the percentage of active compounds was calculated by dividing the number of validated hit compounds (based on a z-score cut-off below -2 or above 2) by the total number of screened compounds and multiplying by 100%. This is now described in the main manuscript on page 7, lines 204-206.

b. **The authors build a training set that is used to generate a predictive model, which then identifies new hits from the testing set data. But there is no explanation for the criteria (e.g. physical and chemical properties) that were employed for the matching of similar compounds.**

RESPONSE: We agree again. We have now provided more details on the creation of the training and the test sets. The training set consisted of the LOPAC compounds and H-89. The details on the linking are now provided in the Supplementary Information (page 4 lines 113-140). The test compounds were all compounds from the PubChem repository. We do not make predictions of compound activity (to identify new hits) based on the similarity of PubChem compounds (in terms of physical and chemical properties) to hits from the LOPAC screen. Rather, we learn to predict the activity of a compound from its target profile. We make such predictions for all PubChem compounds and select those with the highest predicted activity as hits. This is now described in more detail in the Supplementary Information on pages 4-5, lines 113-171.

c. **How was the Z-score and reliability score calculated? And how does the Z-score link with the bacterial load and cell viability?**

RESPONSE: Z-scores are normalized values, obtained by subtracting the mean of the negative control from each individual measurement and dividing by the standard deviation of the negative control. A formula has now been added in the Supplementary Information (page 7, lines 222-226). For the screens, z-scores were calculated separately for bacterial load and cell viability. A z-score of -2 for the latter means that cell viability is 2 standard deviations lower than the average of the negative control. The predictive models we learn also predict z-scores and not actual values. Next to the details on the calculations of the predictions for the z-scores, details on the calculations of the reliability scores are now

provided in the Supplementary Information (page 5, lines 172-180). We now also better distinguish the screening z-scores from the predicted z-scores. The screening z-scores directly relate to the bacterial load or viability of host cells as described above. The predicted z-scores have the same relationship to the predicted bacterial load or cell viability.

- d. **Multiple predictive models were constructed using different bootstrap replicates of the training data set, and the average of different bootstrap replicates were applied for overall prediction. But what are these replicates? An adequate explanation of this is wanting.**

RESPONSE: The procedure is now better explained in the Supplementary Information on page 5, lines 162-171. In brief, bootstrap replicates are not biological or technical replicates, but rather sampled variants of the training dataset. A bootstrap replicate is also called a bootstrap sample, and using this term would probably avoid the confusion. Bootstrap sampling is a standard statistical procedure. A bootstrap sample is obtained by randomly sampling training instances, with replacement, from the original training set, until an equal number of instances as in the training set is included in the sample. Bagging constructs the multiple predictive models by making bootstrap samples of the training set and using each of these samples to construct a predictive model.

8. **Finally, at least in the view of this reviewer, the manuscript is replete with experimental redundancies. For instance, in developing the predictive tool, the entire LOPAC library was matched with PubChem to identify 1058 protein targets. The rationale for this is puzzling. Why not just match the hits from the LOPAC screen?**

RESPONSE: The rationale behind this has now been more clearly described by including a more extensive description of machine learning methods in the Supplementary Information (page 4, lines 113-140). We use the entire screening dataset to learn models for predicting compound activity. As explained under 7b above, we do not predict hits based on the similarity to hits from the screen: this would correspond to using the nearest neighbour prediction method. We rather use tree ensembles that have better predictive power. We make more fine-grained predictions of activity: to make as fine-grained predictions as possible, we use all the targets and not only those of the hits from the LOPAC screen. Also experimentally proven “negative” information could be used in this way to refine the predictive efficacy.

And at the end of this complicated exercise the authors end up with a very small number of predicted hits: 9 for *Mtb* and 4 for *Stm*. This small number is surprising because a simple virtual screen of the 460,580 compounds against a single target protein should have yielded more hits depending on the binding energy cut-off used.

RESPONSE: The updated text on the machine learning methods and selection of candidate compounds in the Supplementary Information (page 4-5, lines 113-161) we hope now explains the limited number of hits. We did not consider single targets when probing the PubChem database but rather used target profiles learned from the LOPAC screening data. As this limits the number of candidate compounds to those affecting a certain target or combination of targets without affecting an array of other targets, a small number of compounds was predicted. **The last segment where a human kinome RNAi screen was used to validate the handful of predicted hits is again another case of needless redundancy. Was an entire kinome scan really necessary? And if the only conclusion from this was that some RTKs are important for**

intracellular bacterial survival, does it add any value given the extensive information already available in the literature

RESPONSE: We are inclined to have a slightly different view than the reviewer regarding the redundancy of the kinome RNAi screen. Taking an “unbiased” RNAi kinome screen approach agnostic to which candidates might be relevant, the entire data set could now be mined and used to independently confirm and validate a RTK signaling network as a prominent host pathway regulating *Mtb*. We would argue that the added value of the RNAi kinome screen did not lie in the confirmation of three targets but rather in the independent confirmation and validation of the compound target testing results.

Reviewer #2

- 1. The hella cells are good model for Salmonella infection assay (salmonella infects epithelial cells) but the melanoma derived cells are poor model for Mtb infection. The data demonstrated the poor performance of this model in the case of Mtb. All the compounds (and not only selected ones) should be tested on monocytes derived or primary phagocytotic cells infected with Mtb and the results should be confirmed in terms of CFU's counts from.**

RESPONSE: For our reply to this comment, please see our response to point 1 from reviewer #1.

- 2. The efficacy results are presented in Z' score values. While these can demonstrate that the assay is robust and reproducible it does not provide a comparable or reliable efficacy measure. Intracellular MIC's or (because of the poor activity of the compounds) % inhibition would be a better measure.**

RESPONSE: For our reply to this comment, please see our response to point 3 from reviewer #1.

- 3. I could not understand the rational for expressing the activity against H-89. Are there issues with independent assessment of the standalone activity of each compound? H-89 can be used as a control but the description of the results by means of the efficacy assay should be better explained.**

RESPONSE: We thank the reviewer for pointing this out and have adjusted the main text (page 5, lines 120-131) to better explain the rationale behind this choice. H89 was our initial lead compound that we reported in our cited Kujil *et al* (Nature 2007) paper. Here we are searching for compounds outperforming H89, as none of the HDT compounds reported in literature that we tested surpassed H89 in our infection model. We have moved the key figure demonstrating comparative data of literature compounds vs. H89 from the Supplementary Information to the main manuscript (new Figure 1). Since we want new candidate compounds for HDT to outperform H-89 in our screens we used the efficacy of H-89 as an additional hit selection criterion beyond the (arbitrary) z-score cut-off at $z=2$ or $z=-2$.

- 4. As the results used a single concentration of inhibitors each of the compound should be verified using dose and time dependent experiments.**

RESPONSE: We fully agree with the reviewer that dose and time titrations should be performed before moving forward towards clinical application (see also reply to reviewer #1, point 3). The focus of our current manuscript was screening and developing a predictive model, for which we chose a single (10 μ M) concentration, which is common practice in this type of screening approaches. In current translational medicine models, which include animal experimentation in

zebra fish and mouse TB models, we will be testing various compound doses and durations of administration.

- 5. The effect of all compounds and SiRNA clones should be controlled to rule out effect on cells metabolic ability to reflect the fluorescent marker (usually done on beads coated with fluorescence protein).**

RESPONSE: We understand the reviewer's concern about metabolic turnover of the fluorescent markers, and agree with this. This was exactly our rationale to confirm all hits in a completely independent, non-fluorescent bacterial colony forming unit (CFU) assay. Thus, any effects on fluorescent marker turnover could be ruled out.

- 6. A better literature search should have been performed. I found that bromo- and Haloperidols were identified already as antimycobacterial agents working in alone and in synergy with antibiotics.**

RESPONSE: We thank the reviewer for pointing this out. We were indeed aware of this finding as we mentioned in the Discussion section, citing Sundaramurthy *et al.* who previously identified haloperidol as a host-directed compound inhibiting intracellular mycobacteria. In our study we used the "identification" of Haloperidol in our screens as a further validation of our approach (next to the data shown in Figure 1). This is the reason why we did not put much emphasis on discussing the haloperidol effect as a finding.

- 7. I found the mix of Salmonella and Mtb experiments confusing and difficult to assess. I think it would be more appropriate to provide a linear description of each one of the organisms and then compare the two.**

RESPONSE: We thank the reviewer for pointing this out. We have now reorganised the manuscript, and have adjusted the sequence of the presentation of the key results accordingly.

- 8. A more appropriate term for your assay is medium throughput.**

RESPONSE: We thank the reviewer for pointing this out, and have adjusted this term throughout the manuscript accordingly.

- 9. The term high content is used in appropriately. High content usually involve assessment of multiple parameters such as expression of proteins, viability toxicity etc. simple measurement of fluorescence is not high content.**

RESPONSE: We thank the reviewer for pointing this out and have adjusted the term throughout the manuscript accordingly.

- 10. The term algorithm is usually associated with flow charts or disclosure of the program source code. From my understanding the process that has been performed is manual bioinformatics analysis of multiple parameters.**

RESPONSE: For our reply to this comment, please see our response to point 5 from reviewer #1.

- 11. A better description of the novelty of the finding should be provided. After all screening of signaling inhibitors and RNAi against intracellular MTb have been performed in the past as well repurposing approach to test for MTb inhibitors.**

RESPONSE: We agree with the reviewer also at this point. We have now provided a broader context to, and discussion of our new findings. We also have fitted our new results now to the top enriched KEGG pathway, identifying neurotrophin signaling as a host *Mtb*-regulating pathway and a target of Dovitinib, which has not been identified in this context before. This was independently confirmed also by silencing Neurotrophic Receptor Tyrosine Kinase 1 (NTRK1). In addition to the drug repurposing screens (which we agree are somewhat similar in methodology to other studies), the newly developed *in silico* prediction model is a major innovative aspect of this manuscript. As this model can easily be adapted to other chemical screens, we are convinced that it is of interest to a broader readership and therefore we believe reporting this methodology is of great interest.

12. **As this paper describes medicinal chemistry approach for drug discovery I encourage the authors to use medicinal chemistry jargon and terms or relate their units (z' score and absorbable).**

RESPONSE: we thank the reviewer for this comment. In the literature we have found the use of z-scores to be rather common when reporting data from large scale chemical or genetic screens. For that reason, and to optimise comparability to such other studies, we would think it would be helpful to leave this in the manuscript in the sections describing the screening results. Furthermore, we have changed absorbance to % of control where relevant.

13. **The term druggable should be better explained. As all compounds only slightly effect mycobacterial growth how could they be considered drugs?**

RESPONSE: In this manuscript 'druggable' refers to protein targets that can be modulated using existing compounds. As the compound-target interactions were taken from PubChem bioassays, these interactions have already been discovered and described by others. For further discussion whether the compounds can be considered drugs, please see our reply to reviewer #1 point 3.

Reviewers' comments:

Reviewer #2 (Remarks to the Author):

Thank you for the revised manuscript. It is much improved now as many of the technicalities were better explained. Yet, several crucial issue remained un-answered and make the conclusions somehow speculative.

1. The efficacy results are presented in Z' score values. While these are commonly used in the industry, when antimicrobial agents or agents that effect microbial clearance are considered, a comparable or reliable efficacy measure should be added to the study. . Intracellular MIC's or (because of the poor activity of the compounds) % inhibition should be provided and ranking of the targets/ compounds should be analyzed based upon the ability to effect intracellular growth. To prove my point, Figure 2C actually showed that all compounds are not effective at all against Mtb, yet the whole experiment is still presented. The same should be performed for the virtual screens and RNAi assays.

This point also highlight key differences between salmonella and Mtb. Its not only the host that is effecting the bacterial growth its also the innate ability of the compounds to kill these two organisms in different manner.

Combining the in silico predictive model and the actual screening and RNAi approach may strengthen the rational of targeting host proteins but it is more like handwaving rather than logical approach. If the output of all approaches (in silico, Z-score screening and RNAi) would be tested in a functional assay (MIC of bacteria in cells) then it is acceptable for publication as it stands right now this is a preliminary excersize that need validation.

2. Another key issue is the problem of mixing known antibiotics and signalling inhibitors in one screen that might skew the results. I would remove known antimicrobial compounds from the screening process.

3. Unfortunately, too many of the findings are validations of previous studies, and might be better fit the supplementary material (fig 1 for example). The authors should highlight their novel findings.

Response to the second round comments of Reviewer #2 from 14th August 2017.

General response to these comments, followed by a point to point reply to specific comments.

We have carefully studied these comments and fear there is a significant misunderstanding on the reviewer's side, which seems to have led to erroneous conclusions and which –understandably in that case- has clearly dampened his/her enthusiasm for the paper. In fact the reviewer states that the paper is acceptable for publication IF “the output of all approaches (in silico, Z-score screening and RNAi) would be tested in a functional assay (MIC of bacteria in cells)”. But this is exactly what we have done and report in the manuscript: all data presented in Figures 1, 2, 3, and 4B, Tables 1, 2 and 3, Suppl. Figures S1, S2, S4 and S6D and Suppl. Tables 2 and 3 are from the functional assays the reviewer is asking for. We are afraid that the reviewer has been assuming that these data were all virtual data, derived from a prediction model, which we only describe later on (Suppl. Tables 5 and 6) but this is not the case at all. Rather the opposite is true, in that the prediction model as illustrated in Figure 3A and Suppl. Figure 5 is based on the functional data presented in Figures 2A and 2D. In fact we even validated all hits in two different and independent functional assays: a fluorescence-based assay described in our manuscript, as well as the classic gold standard bacterial colony forming unit assay, which has been used for decades to measure bacterial growth. Thus we have exactly done what the reviewer suggests, which is that all our findings are based on functional bacterial inhibition assays.

Below we address all comments specifically in a point by point fashion.

1. ***“The efficacy results are presented in Z' score values. While these are commonly used in the industry, when antimicrobial agents or agents that effect microbial clearance are considered, a comparable or reliable efficacy measure should be added to the study. Intracellular MIC's or (because of the poor activity of the compounds) % inhibition should be provided and ranking of the targets/ compounds should be analyzed based upon the ability to effect intracellular growth. To prove my point, Figure 2C actually showed that all compounds are not effective at all against Mtb, yet the whole experiment is still presented. The same should be performed for the virtual screens and RNAi assays.”***

RESPONSE:

The reviewer wants to see, next to the z-score values, the actual % inhibition values to be able to relate the z-scores to the actual extent of the inhibitory effect. We show such Figures now below (for the key validation experiments), side-by-side to the original z-score Figures for comparison: the original z-score data from Figure 2 are shown on the left, the corresponding % inhibition from the same experiments as % of control on the right. These data are essentially identical. We have done the same for the data from Figure 3 (see bottom part of the Figure), and again, these data are essentially identical. This new information proves that the novel compounds we have identified have a strong inhibitory effect on bacterial growth in human cells. (The control Figure 2C shows that these compounds do not act directly on bacteria in classic bacterial cultures, but only act on intracellular bacteria via the host cell, see also below). In Figures 1 and 3 (C,D,E,G,H) we in fact already showed % inhibition values, expressed as % of control of the untreated test conditions, in response to the reviewer's first round of comments. We have modified Figures 2B, 2E, 3B and 3F very significantly, and now shown side-by-side the z scores as well as the percentage inhibition data, expressed as % control. We also describe this now in the text much more clearly (pages 5-6 lines 148-152, 6 lines 161-169, 7 lines 206-208 and

217-220 and 8 lines 224-225), to better highlight the very clear and significant effects of our new compounds on intra- (but not extra-) cellular *Mtb* and *Stm*, indicating that these anti-microbial compounds act via host and not bacterial mechanisms. Thus these compounds are clear and novel candidates for HDT. Furthermore these % inhibition data clearly demonstrate the convincing *extent* of the inhibitory effects of our compounds on bacteria in infected cells.

Figure 2 Comparison of z-score and % of control

Mtb

Stm

Note: Compounds T and O appeared to be known antibiotics, not acting via host mechanisms while all other compounds had no effect on bacteria outside host cells (Figure 2F in the manuscript)

Figure 3 Comparison of z-score and % of control

Mtb

Stm

We have now included these new Figures in the manuscript to clarify these important points and provide the requested novel information (in Figures 2B and 2E and Figures 3B and 3F).

Finally we would like to briefly mention that in several Figures (3C and 3D) in the original manuscript the inhibitory effects of the tested compounds on intracellular *Mtb* or *Stm* were already expressed as % of control values, thus showing exact % inhibition values rather than z scores, the latter of which were only used to plot the data from the initial large screens shown in Figure 2A, 2D, 3B, 3F and 4B.

“To prove my point, Figure 2C actually showed that all compounds are not effective at all against Mtb, yet the whole experiment is still presented. ”

RESPONSE:

The comments on Figure 2C seem to illustrate another misunderstanding. If we understand the reviewer correctly he/she seems to think that this Figure suggests the compounds have no effect on bacteria. Instead this Figure is a negative control Figure, demonstrating that our compounds have no inhibitory effect on *Mtb* or *Salmonella* outside cells. Thus the “negative” results are key to confirm the point that they act on intracellular bacteria via host mechanisms.

“This point also highlight key differences between salmonella and Mtb. Its not only the host that is effecting the bacterial growth its also the innate ability of the compounds to kill these two organisms in different manner.”

RESPONSE:

The reviewer also points out the difference between *Salmonella* and *Mtb*. While this is correct, as we highlighted in Figure 2G, we would like to emphasize that this is one of our important conclusions in the paper, rather than an experimental problem (as the reviewer may think, if we understand him/her correctly). Based on these data we concluded that the host networks involved in controlling the two intracellular pathogens display similarities as well as differences as some compounds work on only one pathogen, while others work on both (this paper and ms. in preparation). This is entirely expected and agrees with the difference in the cell biological behaviour of these vacuolar pathogens, as we discuss in the paper in more detail (page 4 lines-98-101, page 6 lines 171-174 and page 11 lines 331-336). Moreover, this suggests that our strategy can discriminate (likely subtle) cell biological differences between different pathogens based on their susceptibility to different host directed compounds treatments.

“Combining the in silico predictive model and the actual screening and RNAi approach may strengthen the rational of targeting host proteins but it is more like handwaving rather than logical approach. If the output of all approaches (in silico, Z-score screening and RNAi) would be tested in a functional assay (MIC of bacteria in cells) then it is acceptable for publication as it stands right now this is a preliminary excersize that need validation.”

RESPONSE:

Please see our general response to the second round comments of reviewer #2 above.

2. ***“Another key issue is the problem of mixing known antibiotics and signalling inhibitors in one screen that might skew the results. I would remove known antimicrobial compounds from the screening process.”***

RESPONSE:

We were not fully sure whether we read this comment of the reviewer correctly, and considered two possibilities: if the reviewer might have been concerned that we experimentally combined known antibiotics with host signalling inhibitors we would like to mention that this was nowhere the case. We only use antibiotics as positive control to show that we see bacterial inhibition by classic antibiotics (Rifampicin for *Mtb*, gentamycin for *Salmonella*), as e.g. in Figures 2C and 2F: as expected, in these (extracellular) experiments the host directed compounds show no effect. If, however, the reviewer meant that he/she was confused by the finding in Figure 2F, where we see hits for *Salmonella* with compounds T and O, we would like to point out that these 2 compounds (which are known antibiotics for *Salmonella*) were an integral part of the LOPAC library used, and were therefore included in our screens. The fact that these known antibiotics for *Salmonella* are hits in our screen only confirms the strength and validity of our approach, and shows that we can clearly distinguish antibiotics from host-directed compounds.

In addition and importantly, the prediction model we describe is based solely on the human targets of the compounds studied, and not on possible bacterial targets. Thus, the prediction model is not confounded by mixing antibiotics and signalling inhibitors, in case this might have been a concern to the reviewer.

3. ***“Unfortunately, too many of the findings are validations of previous studies, and might be better fit the supplementary material (fig 1 for example). The authors should highlight their novel findings.”***

RESPONSE:

We were somewhat surprised by the comment on Figure 1, because this Figure was indeed supplemental in the initial version but in response to the first round of this reviewers' comments was transferred to the main Figures. More importantly, it is true that this Figure displays “validations of previous studies”, but this is exactly the point of this Figure: it is shown to prove that our novel assays faithfully reproduce previously published hits. This lends important further plausibility to the finding of our new compounds with enhanced bioactivity compared to previous ones, including against multi-drug resistant *Mtb*, a major global health threat.

Response to the original comments of Reviewer #1 from 11th January 2017

1. ***“The reason for use of a melanoma cell line for *Mtb* infection in the screen is not clear. While these cells may be capable of phagocytosing *Mtb*, the question of whether the intracellular environment experienced by the bug is similar to that it would counter within a macrophage has not been clarified. While macrophages would respond to an intracellular infection through an array of anti-bacterial responses, it is unlikely that melanoma cells will be anywhere as aggressive. But surely this would be an important consideration when developing an assay for host-directed drugs? A similar argument can be made for the use of HeLa cells as an infection model for *Stm*.”***

RESPONSE: We understand the reviewer's initial reservation regarding the use of a melanoma cell line as a model for *Mtb* infection. Before better explaining the choice for the MelJuSo and HeLa human cell lines as models, we would like to emphasize that all our compound hits are validated in primary human macrophages as a key gating criterion. Thus, even though cell lines are used for discovery of candidate target molecules and circuits, only those that validated in primary human macrophages qualify as real hits in our study (as exemplified in Figure 3C and Supplementary Figure 1G).

In search for a suitable human cell line model for *Mtb* infection we initially tested several cell lines for potential use in medium throughput chemical genetic screens. The often-used THP-1 human monocytic cell line, however, has the major disadvantage in that it requires PMA stimulation to induce a macrophage-like, adherent phenotype, which profoundly alters PKC signaling. Screening compounds and siRNA libraries on such a PMA-modulated background would complicate the identification of target proteins, and preclude study of molecules in the PKC pathway. In contrast to THP-1, the well documented MelJuSo cell line (e.g. Cell 145:268-283) does not require such additional stimuli. Moreover, it is transfectable with siRNA with high efficiency, is highly homogeneous and has phagocytic capacity. We have shown in the past that human melanocytes are efficient antigen presenting cells (APC) that process and present mycobacteria to CD4 T cells in a HLA class II restricted fashion (J. Immunol.151:7284). We have also worked with MelJuSo as APC in the past to dissect molecular pathways of HLA class II presentation, identifying the role of HLA-DM and HLA-DO (e.g. Current Biology 7:950-957; J ExpMed 191:1127). For these reasons, we decided to develop MelJuSo as a novel human intracellular model for *Mtb* infection, allowing medium-throughput chemical and genetic screens.

Figure 1 now shows important data that validate the new MelJuSo model by reproducing the *Mtb*-inhibiting effect of already published host-directed compounds. Perhaps we made insufficiently clear in the manuscript that this is the key message of Figure 1, for which we apologize. We have therefore amended the text to avoid any potential confusion (page 5, lines 124-127 and 133-136). In the first round revised manuscript we had already moved this Figure from the Supplementary Information to the main manuscript to stress this important validation data, as they underpin the relevance of this human cell based screening system. The added text now better explains the rationale behind the use of the MelJuSo cell line and its advantages as a new human model for chemical genetic screens (page 4 lines 92-97 and page 5 lines 123-136 in the main manuscript and page 1 lines 12-15 in the Supplementary Information). To further underpin its utility, we now provide new data showing that MelJuSo cells are equipped with important antimicrobial host defense mechanisms as judged by the expression of key genes coding for antimicrobial effectors and regulators, based on existing microarrays (courtesy of

Prof. Dr. J.J. Neefjes, LUMC, The Netherlands; confidential, personal communication). We have attached the raw expression data as a table below.

[Redacted]

- 2. “Continuing on the assays, Suppl Fig 1A shows that only about 5%, or less, of the HeLa cells were infected with Stm under the conditions employed. Similarly, the fraction of melanoma cell infected with Mtb was of the order of about 30%. It is doubtful if such low infection loads would yield sensitive and reliable readouts. Indeed the correlation plots shown in Suppl Fig 1F-E&G are in fact very poor with points just bunching at either end rather than displaying a true linear relationship.”***

RESPONSE: We understand the reviewer's concerns regarding the infection rates. Using an identical *Mtb* infection protocol in primary human macrophages, results routinely are in very

similar ranges of infection rates as observed in MeJJuSo. However, in these human macrophages both *Mtb* and *Stm* infection rates differ between experiments and between donors, ranging from ~5% to ~60%. In other published studies mostly CFU (colony forming units as a measure of the number of bacteria) assays are used, but these provide no indication of the actual percentage of infected cells at all, so comparison of our results with literature is challenging. We have performed extensive *Mtb* and *Stm* MOI titrations in primary macrophages, aiming for the highest possible infection rate without affecting host cell viability, and consistently find that a significant proportion of cells remains uninfected (this is in fact the topic of an independent project).

We agree with the reviewer that Supplementary Figures 1E and 1G were not convincing and that it would have been better to base these on a bacterial titration rather than a limited number of different compound treatments. As these figures were merely a different representation of data readily available in the bar graphs in Supplementary Figure 1, we have therefore removed them from the manuscript.

3. ***“The majority of the screen results are given in terms of z-scores and it is difficult to determine how potent the hits really are. In the one instance where the efficacy data is actually provided (Fig. 2D&E) the effect on Mtb infection is only marginal with a reduction of only about 50% or less in CFU values.”***

RESPONSE: We would like to mention that the use of z-scores is quite common in reporting data from chemical screens (see also reply to point 7c below for further details). In contrast to many other studies on novel compounds targeting intracellular *Mtb* that typically study the effects of compounds over a time window of several days and sometimes add these in multiple successive doses, we instead have decided to test our compounds after only a single overnight treatment. This strategy allows rapid screening and reveals only the most potent compounds. The effects we see on *Mtb* with these compounds have not been optimised further, which will be the focus of further translational medicine studies in our lab, using animal models (zebrafish, mice). Thus our model is extremely stringent and designed to identify only the very best hits; to exemplify this, Imatinib did not pass our strictest selection criteria, despite its reported efficacy *in vivo* (see: Napier *et al.* Cell Host Micr 10, 475–485). We believe that our strategy helps finding compounds that might display superior activity against *Mtb* and *Stm*.

“It is questionable whether such compounds can really be considered as drugs.”

RESPONSE: Despite the incompletely sterilizing effects of individual compounds in our single-dose, short term (overnight) setup, we would contend that our thus defined host-directed compounds can be considered novel drugs as they may be used in conjunction with classical antibiotics to 1) kill residual “dormant/nonreplicating” bacteria that are metabolically inactive and thus do not respond to antibiotics; 2) shorten current lengthy TB treatment regimens (6-9 month regimen); and 3) help treating MDR/XDR-TB resistant to most available drugs. Moreover it is well possible that combinatorial regimens of different classes of HDT compounds may exert superior activity against intracellular bacteria by targeting multiple synergizing host pathways. In cancer, combinatorial immunotherapy is becoming more and more in vogue, using e.g. immune checkpoint inhibitors in combination with kinase inhibitors (e.g. Nature 543:728, 2017).

4. ***“The use of H89 as an Akt inhibitor is puzzling. At least to this reviewer’s knowledge, H89 is a PKA inhibitor. For Akt, more efficient and specific inhibitors (e.g. MK2206) are available.”***

RESPONSE: We indeed agree that H89 has mostly been known as a PKA inhibitor. However, in our previous paper by Kuijl, C., *et al.*, (Intracellular bacterial growth is controlled by a kinase network around PKB/AKT1. *Nature* 450, 725–730; 2007) we have demonstrated that H89 mediated its effects on both *Stm* and *Mtb* inhibition by inhibiting PKB/AKT1 rather than PKA. This was confirmed by using additional chemical inhibitors as well as genetically (RNAi). We have added additional text and the above reference to the main manuscript on page 3 lines 63-67 to explain this.

5. ***“The use of predictive models, a training set and a testing set for the identification of novel hits is claimed to represent a new algorithm. However from all the steps it is clear that this is just a pipeline that links already established methods to identify new hits. Describing this as an algorithm, therefore, is a bit misleading.”***

RESPONSE: We agree with this comment, and have updated the text accordingly, describing our new approach as a predictive model rather than an algorithm.

6. ***“Further, to validate the above protocol, the authors state “Commercially available compounds predicted to strongly decrease bacterial load were selected...” But this statement is not accompanied anywhere by an explanation of how such a selection was done. Was it on the basis of binding of compounds in PubChem to targets for the hits in the LOPAC screen? Even if that is the case, how do they predict that the compounds will decrease bacterial load, without affecting host cell viability?”***

RESPONSE: We fully agree with the reviewer that this section was lacking clarity, and thank the reviewer for pointing this out. We have completely rewritten this section, and now provide a much more detailed explanation of how the *in silico* experiments were performed, both in the main manuscript (pages 6-7, lines 177-203) and in the Supplementary Information (pages 3-6, lines 96-200). We have also updated Figure 3A and Supplementary Figure 5A to further clarify the prediction pipeline and *in silico* methods. The extended explanation of the methods should now also clarify the selection criteria used that resulted in the final set of selected compounds. In brief, we linked all LOPAC compounds to PubChem, and bioassay data were retrieved, identifying 1058 confirmed human protein targets of these 1260 compounds. This resulted in a data table of all LOPAC compounds annotated with their bacterial load and cell viability z-scores from the performed screens, as well as their PubChem bioassay activity. This was then used as a training set to learn ensembles of predictive clustering trees to predict impact on intracellular bacterial survival and host cell viability of compounds in the PubChem repository. Querying PubChem for compounds confirmed to target at least one of the 1058 identified proteins yielded 460,580 compounds, which were then annotated with their bioassay data and fed into the predictive model as a test set to predict their intracellular bacterial load and host cell viability z-scores. Together with the predictions, we calculated a reliability score based on the variance of the predictions in the ensemble. Based on the predicted intracellular bacterial load and host cell viability z-scores, we generated a list of candidate hit compounds. We then further filtered this list of candidates by only considering predictions that are most reliable (using the reliability score) and finally selected compounds that were commercially available for further experiments.

7. ***“The description of how the predictive tool was developed is very confusing, with no rationale provided for many of the steps involved. Some pertinent points here are:”***

a. ***“There is no clarity on how percentage of active compounds was calculated. What values were used?”***

RESPONSE: We agree again with the reviewer that this description was lacking clarity, and thank the reviewer for pointing this out. In answer to point 7a, the percentage of active compounds was calculated by dividing the number of validated hit compounds (based on a z-score cut-off below -2 or above 2) by the total number of screened compounds and multiplying by 100%. This is now described in the main manuscript on page 8, lines 230-233.

b. ***“The authors build a training set that is used to generate a predictive model, which then identifies new hits from the testing set data. But there is no explanation for the criteria (e.g. physical and chemical properties) that were employed for the matching of similar compounds.”***

RESPONSE: We agree again. We have now provided more details on the creation of the training and the test sets. The training set consisted of the LOPAC compounds and H-89. The details on the linking are therefore now provided in the Supplementary Information (page 4 lines 115-142). The test compounds were all compounds from the PubChem repository. We did not make predictions of compound activity (to identify new hits) based on the similarity of PubChem compounds (in terms of physical and chemical properties) to hits from the LOPAC screen, but rather, used machine learning to predict the activity of a compound from its target profile. We made such predictions for all PubChem compounds and selected those with the highest predicted activity as candidate hits for further experimental testing. This is now described in more detail in the Supplementary Information on pages 4-5, lines 114-173.

c. ***“How was the Z-score and reliability score calculated? And how does the Z-score link with the bacterial load and cell viability?”***

RESPONSE: Z-scores are normalized values, obtained by subtracting the mean of the negative control from each individual measurement and dividing by the standard deviation of the negative control. A formula has now been added in the Supplementary Information (page 7, lines 225-229). For the screens, z-scores were calculated separately for intracellular bacterial load and host cell viability. A z-score of -2 for the latter means that cell viability is 2 standard deviations lower than the average of the negative control. The predictive models we learn also predict z-scores and not actual values. Next to the details on the calculations of the predictions for the z-scores, details on the calculations of the reliability scores are now provided in the Supplementary Information (page 5, lines 174-182). We now also better distinguish the screening z-scores from the predicted z-scores. The screening z-scores directly relate to the intracellular bacterial load or viability of host cells as described above. The predicted z-scores have the same relationship to the predicted intracellular bacterial load or host cell viability.

d. ***“Multiple predictive models were constructed using different bootstrap replicates of the training data set, and the average of different bootstrap replicates were applied for overall prediction. But what are these replicates? An adequate explanation of this is wanting.”***

RESPONSE: The procedure is now better explained in the Supplementary Information on page 5, lines 164-173. In brief, bootstrap replicates are not biological or technical replicates, but rather sampled variants of the training dataset. A bootstrap replicate is also called a bootstrap sample, and using this term should avoid the confusion. Bootstrap sampling is a standard statistical procedure. A bootstrap sample is obtained by randomly sampling training instances, with replacement, from the original training set, until an equal number of instances as in the training set is included in the sample. Bagging constructs the multiple predictive models by making bootstrap samples of the training set and using each of these samples to construct a predictive model.

8. ***“Finally, at least in the view of this reviewer, the manuscript is replete with experimental redundancies. For instance, in developing the predictive tool, the entire LOPAC library was matched with PubChem to identify 1058 protein targets. The rationale for this is puzzling. Why not just match the hits from the LOPAC screen?”***

RESPONSE: The rationale behind this has now been more clearly described by including a more extensive description of machine learning methods in the Supplementary Information (page 4, lines 115-142). We use the entire screening dataset to learn models for predicting compound activity. As explained under 7b above, we did not predict hits based on the similarity to hits from the screen: this would correspond to using the nearest neighbour prediction method. We rather used tree ensembles that have better predictive power, and make more fine-grained predictions of activity. To make as fine-grained predictions as possible, we used all the targets and not only those of the hits from the LOPAC screen. Thus, also experimentally proven “negative” information can be used in this way to refine the predictive efficacy.

“And at the end of this complicated exercise the authors end up with a very small number of predicted hits: 9 for Mtb and 4 for Stm. This small number is surprising because a simple virtual screen of the 460,580 compounds against a single target protein should have yielded more hits depending on the binding energy cut-off used.”

RESPONSE: The updated text on the machine learning methods and selection of candidate compounds in the Supplementary Information (page 4-5, lines 114-163) we hope now explains the limited number of hits. We did not consider single targets when probing the PubChem database but rather used target profiles identified from the LOPAC screening data. As this limits the number of candidate compounds to those affecting a certain target or combination of targets without affecting an array of other targets, a small number of compounds was predicted.

“The last segment where a human kinome RNAi screen was used to validate the handful of predicted hits is again another case of needless redundancy. Was an entire kinome scan really necessary? And if the only conclusion from this was that some RTKs are important for intracellular bacterial survival, does it add any value given the extensive information already available in the literature.”

RESPONSE: We are inclined to have a slightly different view than the reviewer regarding the redundancy of the kinome RNAi screen. Taking an “unbiased” RNAi kinome screen approach agnostic to which candidates might be relevant, the entire data set could now be mined and used to independently confirm and validate a RTK signaling network as a prominent host pathway regulating *Mtb*. We would argue that the added value of the RNAi kinome screen did

not lie in the confirmation of three targets but rather in the independent confirmation and validation of the compound target testing results.

REVIEWERS' COMMENTS:

Reviewer #1 (Remarks to the Author):

The revised manuscript is significantly improved over the earlier version. There is much greater clarity now on the logic for using the three separate approaches (chemical library screen, siRNA screen, and the in silico model), as well as the complementarity between them. The development of the in silico model is also better described and is easier to understand. All in all, my queries have been satisfactorily answered.

One minor point: The hits from the LOPAC screen are commonly referred to as 'novel compounds' (e.g. line 41 of the Abstract). This is technically incorrect as it is not the compounds that are novel, but the biological activities identified for them.

Reviewer #2 (Remarks to the Author):

Thanks you for making all the corrections and clarifications. The paper reads well now and is much clearer and easier to understand. Only minor remarks is that that the drug candidates concentration (10uM) should be mentioned in the legends to the figure in addition to the method section.

REVIEWERS' COMMENTS

Reviewer #1 (Remarks to the Author):

The revised manuscript is significantly improved over the earlier version. There is much greater clarity now on the logic for using the three separate approaches (chemical library screen, siRNA screen, and the in silico model), as well as the complementarity between them. The development of the in silico model is also better described and is easier to understand. All in all, my queries have been satisfactorily answered.

One minor point: The hits from the LOPAC screen are commonly referred to as 'novel compounds' (e.g. line 41 of the Abstract). This is technically incorrect as it is not the compounds that are novel, but the biological activities identified for them.

Author response:

We thank the reviewer for pointing out this textual inaccuracy. We have adjusted the manuscript to correct this, in track change mode.

Reviewer #2 (Remarks to the Author):

Thanks you for making all the corrections and clarifications. The paper reads well now and is much clearer and easier to understand. Only minor remarks is that that the drug candidates concentration (10uM) should be mentioned in the legends to the figure in addition to the method section.

Author response:

We appreciate that the reviewer pointed out this small omission. We have included the drug candidate concentrations in the manuscript where relevant, in track change mode.